# Cross-neutralizing and potent human monoclonal antibodies against historical and emerging H5Nx influenza viruses

Alexandra A. Abu-Shmais ®[1,6], Gray Freeman[1,6], Adrian Creanga ®[1], Matthew J. Vukovich[1], Tek Malla[2], Grace E. Mantus[1], Geoffrey D. Shimberg[1], Rebecca A. Gillespie ®[1], Vanessa Guerra Canedo[1], Bernadeta Dadonaite ®[3], Megan D. Rodgers[1], Ankita J. Chopde ®[1], Elizabeth Bardwil-Lugones ®[1], Tatsiana Bylund[1], Amy R. Henry[1], Jesmine Roberts-Torres[1], Timothy S. Johnston[1], Sarah Smith ®[1], Eun Sung Yang[1], Cheng Cheng ®[1], Emma L. Walker[4], Michelle Ravichandran ®[1], Ingelise J. Gordon[1], Tejaswi S. Dittakavi ®[1], Douglas S. Reed ®[4], Theodore C. Pierson[1], Lesia Dropulic ®[1], Jesse D. Bloom ®[3,5], Yaroslav Tsybovsky ®[2], Eli A. Boritz[1], Daniel C. Douek ®[1], Tongqing Zhou ®[1]✉, Masaru Kanekiyo ®[1]✉ & Sarah F. Andrews ®[1]✉

Highly pathogenic avian influenza H5Nx viruses are an emerging threat for global health, especially clade 2.3.4.4b H5N1 virus which causes panzootic infections. Here we describe the isolation and characterization of broadly cross-neutralizing monoclonal antibodies (mAbs) against diverse H5Nx viruses from individuals who received a monovalent H5N1 vaccine 15 years ago. By screening over 500 mAbs, we identified 5 mAbs that neutralized the majority of H5 clades including 2.3.4.4b and target three distinct conserved epitopes within the HA globular head. Cryo-electron microscopy structures of these mAbs in complex with HA, deep mutational scanning and neutralization escape studies define the sites of vulnerability of H5 HA. These mAbs mediated stronger prophylactic protection against clade 2.3.4.4b H5N1 infection in mice than the best-in-class mAb targeting the HA stem. Our study identified several highly potent broadly neutralizing H5 mAbs from humans that either alone or in combination provide a pragmatic pandemic preparedness option against the threat of panzootic H5N1 influenza.

Highly pathogenic avian influenza (HPAI) H5N1 poses a substantial public health threat with pandemic potential. The first H5N1 human fatality was in 1997 in Hong Kong after genetic reassortment between A/goose/Guangdong/1/1996 (Gs/GD lineage) and other low pathogenic avian influenza (LPAI) viruses[1]. Since then, sporadic H5N1 human infections have occurred with a 51% case fatality rate of laboratory confirmed infections[2]. The emergence of clade 2.3.4.4b H5N1 from a 2.3.4.4b H5N8 reassortment in 2020[3] has caused global alarm, as the virus continues to spread rapidly in the Western hemisphere[4]. In particular, detection of clade 2.3.4.4b across diverse hosts, including terrestrial and marine mammals[5–7], demonstrates a concerning shift in the epidemiological landscape of H5N1. Since March 2024, this clade has caused sizeable outbreaks in US dairy farms, the first HPAI 2.3.4.4b H5N1 infections in dairy cattle[8]. Human-to-human transmission has not been observed; however, reports of HPAI H5N1 infections among farm workers with direct contact with infected cattle and poultry[9] underscores the public health risk of this panzootic pathogen. Besides clade 2.3.4.4b H5N1 virus, a recent surge in human infections with clade 2.3.2.1e H5N1 virus in Southeast Asia with a 43% fatality rate in symptomatic individuals[10] further highlight the urgent need for effective medical countermeasures.

**Fig. 1 | Isolation of H5 TX/24 specific antibodies. a**, Schematic diagram of VRC310 clinical trial vaccine schedule. Individuals received either H5 A/Indonesia/05/2005 HA DNA vaccine, followed by subvirion H5N1 monovalent inactivated vaccine (MIV) (A/Indonesia/05/2005), or a two-dose regimen of H5N1 MIV. Blood was collected at 2 weeks p.b. and at 6 months p.b. **b**, Gating strategy for single-cell sorting of H5 TX/24+ memory B cells. Cells that bound H5 Indo/05 ectodomain but not headless H5 stem were further selected for binding H5 TX/24. A subset of cells also bound H1 HA (cocktail of H1 NC/99 and H1 Cal09). RATP-Ig supernatants from five 96-well plates at 2 weeks p.b. and five 96-well plates at 6 months p.b. were screened for H5 HA-specific binding by MSD. Created in BioRender (https://BioRender.com/elo13eu). CK, kappa constant region; pA, polyA tail; RATP-Ig, rapid assembly, transfection and production of immunoglobulins.

Antibody-based biologics could provide protection if available in a timely manner, particularly among populations with historically poor vaccine responsiveness, such as geriatric and immunocompromised individuals, as demonstrated for SARS-CoV-2 (ref. [11]). Most of the protective antibody response to influenza is directed to the viral surface glycoprotein haemagglutinin (HA). HA is a homotrimeric class 1 fusion protein composed of functionally competent HA1 and HA2 subunits. The immunodominant HA1 globular head is responsible for binding to sialic acids on the host cell surface and defines receptor tropism (that is, α2,3- versus α2,6-linked sialic acids), while the immunologically subdominant HA stem, comprising mostly HA2, mediates fusion between viral and endosomal membranes. Potently neutralizing antibodies primarily target the globular head and are characteristically strain specific because of the hypervariable antigenic landscape of the HA head. Neutralizing antibodies against the conserved HA stem region are less frequent; however, they are broadly cross-reactive across HA subtypes, albeit less potent[12,13].

To isolate the most potent and broadly protective monoclonal antibodies (mAbs), we focused our discovery efforts on HA head-specific antibodies. Using samples from a 2010 A/Indonesia/05/2005 (Indo/05) vaccine trial[14], we identified five potently neutralizing mAbs against both historical and emerging H5Nx viruses that target the HA globular head and confer protection in mice against A/dairy cattle/Texas/24008749001/2024 2.3.4.4b H5N1 virus challenge. Our findings highlight the translational potential of these H5 mAbs for further clinical development as pandemic countermeasures.

## Results

### Isolation of H5 HA-specific antibodies

To isolate H5 HA-specific antibodies, we took peripheral blood mononuclear cells (PBMCs) from a Phase I clinical trial[14] 2 weeks or 6 months after two H5 Indo/05 vaccine doses. Isotype-switched B cells that bound the HA ectodomain of the vaccine strain (H5 Indo/05) and clade 2.3.4.4b H5 strain A/Texas/37/2024 (H5 TX/24) but not the headless H5 stem (HA head+) were single-cell sorted into 96-well plates (Fig. 1a,b, and

Extended Data Fig. 1 and Extended Data Table 1). Using the RATP-Ig[15] workflow (Fig. 1b), we obtained supernatants containing secreted mAbs from each single B cell and screened them for binding to H5 TX/24 HA and the ability to neutralize H5N1 TX/24 virus. A total of 501 wells (~60% of sorted B cells) expressed antibodies reactive to H5 TX/24 HA, 282 wells expressed clonally distinct paired immunoglobulin (Ig) sequences and of these, 128 had some neutralizing activity. We recombinantly expressed these 128 Igs and other highly cross-reactive Igs, totalling 135 clonally distinct mAbs. These mAbs had a diverse Ig repertoire and an average of 5% heavy-chain somatic hypermutation (SHM) (Extended Data Fig. 2).

We tested the 135 mAbs for binding to a panel of recombinant HAs from human viral isolates representative of seven historical or current H5 clades, along with two H1 and two H2 strains (Fig. 2a). All mAbs bound H5 TX/24 and H5 Indo/05 (Extended Data Fig. 3a). mAbs isolated from 2 weeks post boost (p.b.) PBMCs were predominately cross-subtype binding and few had neutralizing activity as measured by the 80% inhibitory concentration ($IC_{80}$) (Fig. 2b,c). mAbs isolated from cells at 6 months p.b. were largely H5 specific, with many binding to 6–7 H5 HAs, and had higher mean neutralizing activity (Fig. 2b,c). The mAbs with the highest potency were generally those with less breadth (Fig. 2d).

To map the HA head region targeted by the mAbs, we used H5 TX/24 mutants to identify mAbs that recognized regions surrounding the receptor binding site (RBS) as well as upper and lower lateral regions (Fig. 2e and Extended Data Fig. 3b,c), confirmed by negative stain electron microscopy (nsEM) images. Several mAbs did not bind H5 HA with a glycan at $90_{HA1}$ (Glyc90 sensitive) (Fig. 2e and Extended Data Fig. 3b) and may represent trimer interface-binding mAbs[16,17]. nsEM and competition studies further identified mAbs that bound a region at the base of the HA head (Fig. 2e and Extended Data Fig. 3d,e). Over 60% of the mAbs obtained from cells at 2 weeks p.b. were Glyc90 sensitive, while mAbs from 6 months p.b. evenly targeted the other 4 HA head regions (Fig. 2f). Most mAbs binding around the RBS or upper lateral region had HAI activity against TX/24 H5N1 virus and had the

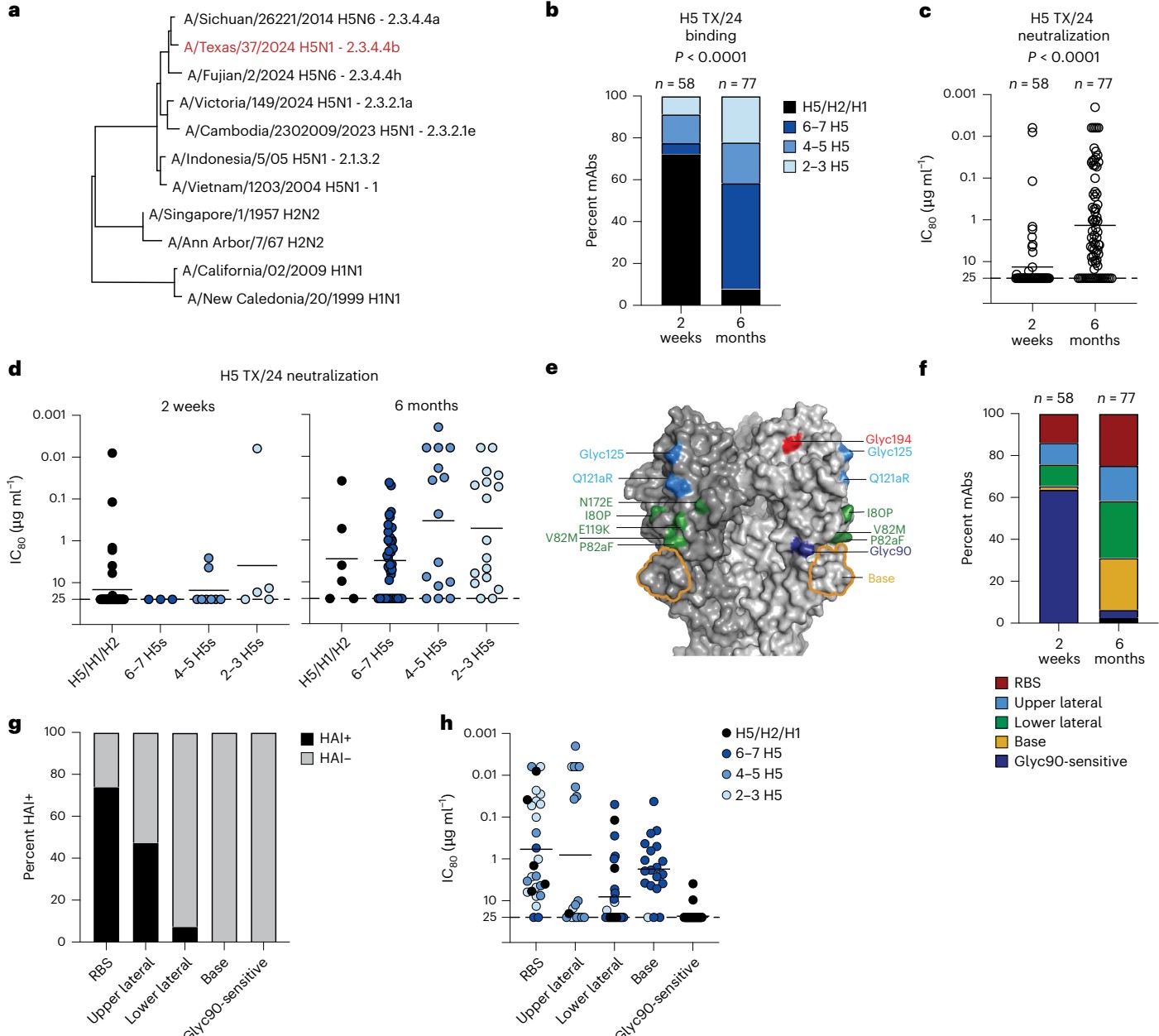

**Fig. 2 | Characterization of H5 TX/24 specific antibodies. a**, Phylogenetic relationship of select H5, H2 and H1 HAs. The H5 GISAID clade is indicated after each virus strain name. H5 TX/24 is in red. **b**, Proportions of recombinantly produced mAbs at 2 weeks p.b. and 6 months p.b. that display specific patterns of binding reactivity: H5/H1/H2 (bind at least one H5 strain plus H1 and/or H2), or cross-reactivity within the H5 subtype to different numbers of H5 HA strains as indicated. Total number of mAbs tested is indicated above the graph. **c**, Neutralization $IC_{80}$ of mAbs at 2 weeks p.b. and 6 months p.b. to H5N1 TX/24. **d**, Neutralization $IC_{80}$ of mAbs at either 2 weeks p.b. (left) or 6 months p.b. (right) differentiated by binding cross-reactivity as in **b**. In **c** and **d**, each dot represents one mAb, with line indicating the mean. **e**, Representation of HA mutations used to differentiate epitope regions of mAbs, colour coded to match HA

region designated in **f**. Glyc, mutations that introduced a glycan at the indicated position. The outlined HA base region was identified through nsEM and/or competition experiments (Extended Data Fig. 3). **f**, Percentage of mAbs from each timepoint that target each of the indicated HA regions, colour coded to match mutations in **e**. **g**, Percentage of mAbs targeting each region that inhibited red blood cell haemagglutination (HAI+) by H5N1 TX/24 virus at a concentration of 25 µg ml[-1] or lower. **h**, Neutralization $IC_{80}$ of mAbs differentiated by binding region. Each dot represents one mAb colour coded by cross-reactivity as in **b**, with the line indicating the mean. The dashed line in **c**, **d** and **h** represents the highest mAb concentration tested. Statistical significance in **b** by Fisher's exact test, in **c** by Mann–Whitney test. All statistical tests were two-sided.

highest neutralization potency but lower overall breadth (Fig. 2g,h). Eight mAbs binding the upper lateral epitope had particularly high neutralizing ability (Fig. 2h) and all eight, from multiple individuals, were encoded by kappa chain variable gene (IGKV) 2D-28, without an apparent heavy variable gene (IGHV) enrichment, suggesting a light-chain-mediated convergence of B cells targeting this region (Extended Data Fig. 3f). Altogether, vaccination with H5 Indo/05

elicits B cells expressing mAbs cross-reactive to H5 TX/24 that target a range of HA head epitopes with varying levels of neutralizing potency and breadth across H5Nx clades.

### Five broad and potent H5 mAbs
Despite the trend of decreasing neutralization potency with increased breadth, two RBS-targeting and one upper lateral-targeting mAbs,

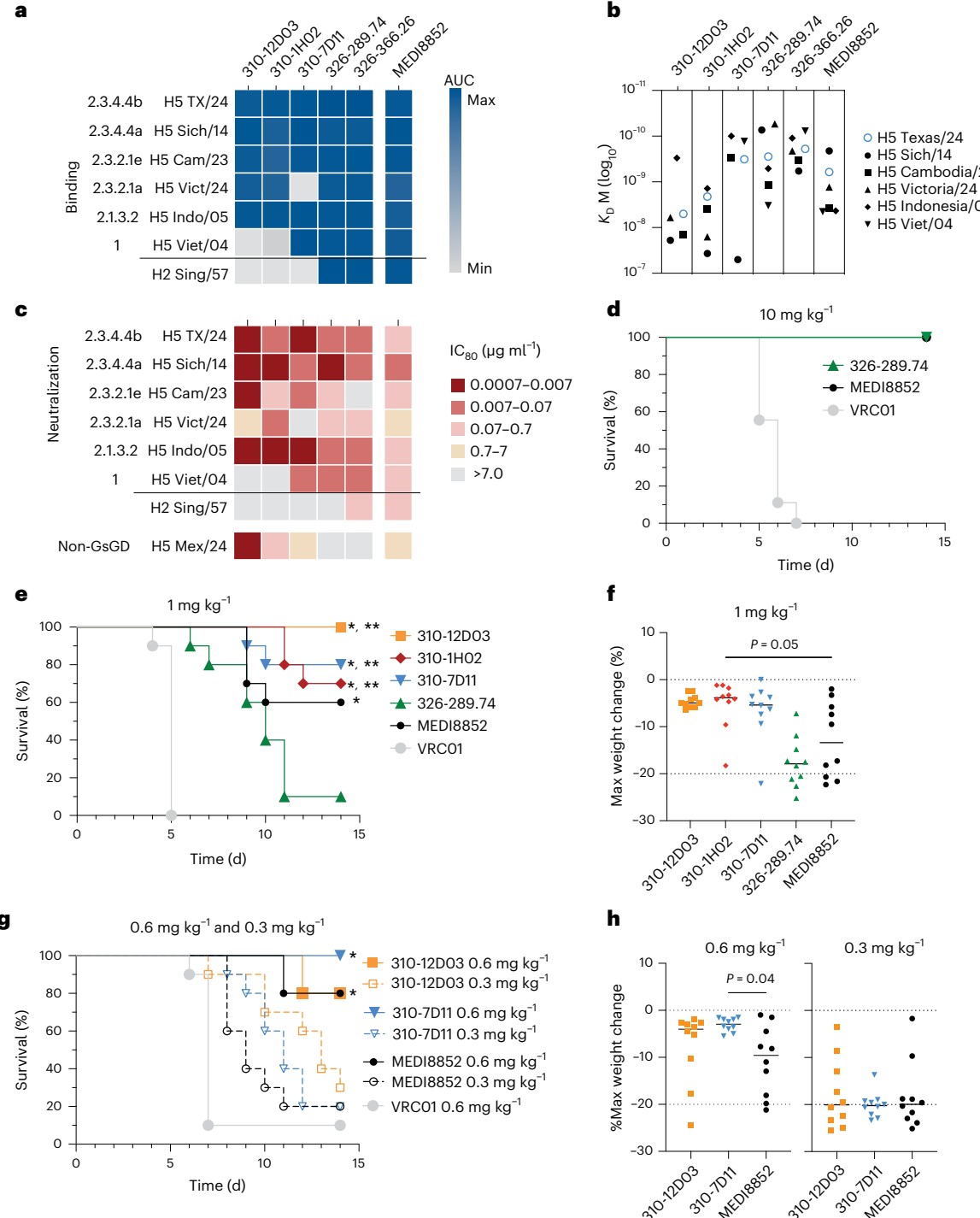

**Fig. 3 | Characterization of top mAbs. a**, Binding of top candidates and MEDI8852 by MSD to indicated H5 HA strains, displayed as a heat map of the AUC (area under the binding curve), with minimum AUC displayed in grey and maximum AUC shown in blue. Data representative of at least 2 independent experiments. **b**, Equilibrium dissociation constants $K_D$ (M) of lead candidates and MEDI8852 by SPR as indicated. **c**, Neutralization $IC_{80}$ of top mAbs and MEDI8852 against a panel of circulating and historical H5 and H2 strains. $IC_{80}$ values displayed as a heat map, with no neutralization shown in grey and increasing potency shown in darker shades of red as indicated. Data representative of 3 or more independent experiments. **d**, Kaplan–Meier survival of BALB/c mice ($n = 8–9$ per group) infected with H5 TX/24 after passive transfer of MEDI8852 and 326-289.74 at 10 mg ml$^{-1}$. **e,f**, Kaplan–Meier survival of BALB/c mice ($n = 10$ per group) infected with H5 TX/24 after passive transfer of mAbs at 1 mg kg$^{-1}$ (**e**) and maximum weight loss (**f**). **g,h**, Kaplan–Meier survival of BALB/c mice ($n = 10$ per group) infected with H5

TX/24 after passive transfer of 310-12D03, 310-7D11 and MEDI8852 at 0.6 and 0.3 mg kg$^{-1}$ (**g**) and maximum weight loss (**h**). In **f** and **h**, horizontal lines indicate group mean. Upper and lower dashed lines indicate no change (0%) and humane euthanasia threshold (−20%) for the study. For **e** and **g**, statistical significance of survival was determined by log-rank (Mantel–Cox) test with Bonferroni–Sidak adjustment. *$P < 0.001$ significant relative to VRC01, **$P < 0.03$ significant relative to 326-289.74. For comparisons against VRC01: $P < 0.0001$ for 310-12D03, 310-1H02, 310-7D11, 326-289.74 and MEDI8852 (**e**), and $P = 0.0004$ for 310-12D03 at 0.6 mg kg$^{-1}$, $P = 0.0001$ for 310-7D11 at 0.6 mg kg$^{-1}$ and $P = 0.0004$ for MEDI8852 at 0.6 mg kg$^{-1}$ (**g**). For comparisons against 326-289.74: $P < 0.0001$ for 310-12D03, $P = 0.0032$ for 310-7D11 and $P = 0.0011$ for 310-1H02 (**e**). Virus inoculum used in the challenge was titrated in a plaque assay to confirm the dose given to mice. Statistics in **f** and **h** by Kruskal–Wallis test with Dunn's multiple comparisons test. Min, minimum; Max, maximum.

termed 310-12D03, 310-1H02 and 310-7D11, respectively, were able to bind five or more H5 HA clades and had a neutralization $IC_{80}$ to H5 TX/24 of 0.02 µg ml$^{-1}$ or less. In addition, we identified two mAbs, 326-289.74 and 326-366.24, with this breadth–potency criterion, from a participant in the 2010 H5 vaccine study obtained from PBMCs isolated 1 week after the participant received an additional experimental vaccine (NCT05968989) containing H2 A/Singapore/1/1957 (Sing/57) HA 13 years later (Extended Data Fig. 4). Each of the five mAbs were encoded by different IGHV and IGK/LV genes and exhibited a range of heavy and light-chain SHMs (Extended Data Table 2). All five mAbs were able to bind H5 Indo/05 and TX/24, as well as A/Sichuan/26221/2014 (Sich/14) and A/Cambodia/2302009/2023 (Cam/23), with apparent affinities ranging from $1.4 \times 10^{-8}$ M to $6.1 \times 10^{-10}$ M (Fig. 3a,b and Extended Data Fig. 5). They also neutralized TX/24 H5N1, Sich/14 H5N6, Indo/05 H5N1 and Cam/23 H5N1 virus, except for one mAb that did not neutralize Cam/23 H5N1 virus (Fig. 3c). 310-12D03 and 310-7D11 had the greatest potency against the currently circulating TX/24 and Cam/23, with $IC_{80}$ of 0.01 µg ml$^{-1}$ or lower for both H5N1 strains (Fig. 3c and Extended Data Table 2). In addition, 310-12D03 potently neutralized a non-Gs/GD H5H2 virus from a recent human fatal case in Mexico[18] (Fig. 3c). In comparison, HA stem mAb MEDI8852 that has been evaluated in a clinical trial for efficacy against seasonal influenza viruses[19,20], had broad binding and neutralization of all the H5Nx and H2N2 viruses tested, albeit at a ≥10-fold lower potency compared with the HA head-specific mAbs for most strains (Fig. 3a–c). Thus, we identified H5-head-specific mAbs that neutralized both clade 2.3.4.4b and 2.3.2.1e H5N1 viruses with higher potency than the best-in-class HA stem-directed mAb.

## H5 mAbs protect against H5N1 lethal challenge

We then tested the ability of mAbs to protect against bovine TX/24 H5N1 infection in mice 24 h after administering either anti-H5 mAbs or control anti-HIV-1 mAb VRC01 (ref. 21). This virus is highly pathogenic in mice and causes neurological symptoms[22,23]. We first tested MEDI8852 and 326-289.74 at 10 mg kg$^{-1}$ and saw that they conferred full protection in vivo (Fig. 3d and Extended Data Fig. 6a). To stratify the mAbs by level of protection, we tested MEDI8852, 326-289.74, 310-12D03, 310-1H02 and 310-7D11 at 1.0 mg kg$^{-1}$. Mice given 310-12D03, 310-7D11 and 310-1H02 were more protected than those that received 326-289.74 (100%, 80% and 70% survival, respectively), with less than 5% weight loss, while MEDI8852 resulted in 60% survival with more pronounced weight loss (Fig. 3e,f and Extended Data Fig. 6b). In addition, we tested 310-12D03, 310-7D11 and MEDI8852 at 0.6 mg kg$^{-1}$ and 0.3 mg kg$^{-1}$. All three mAbs at 0.6 m kg$^{-1}$ conferred 80% or more survival, but mice that received MEDI8852 had greater weight loss (Fig. 3g,h and Extended Data Fig. 6b). Mice given 0.3 mg kg$^{-1}$ of mAb suffered substantial weight loss although they had extended survival over the control VRC01 by 2–6 days (Fig. 3g,h and Extended Data Fig. 6b). Thus, antibodies with the greatest neutralization potency, 310-12D03 and 310-7D11, provided the greatest protection against H5N1 infection at 1.0 and 0.6 mg kg$^{-1}$. In addition, pharmacokinetics studies of these mAbs in a humanized mouse model showed comparable serum half-lives to anti-HIV-1 mAb VRC01LS used in human clinical trials[24], suggesting that they would be viable candidates for prophylaxis treatment in humans (Extended Data Fig. 6d).

## mAbs target distinct regions on the HA head

Cryogenic electron microscopy (cryo-EM) of H5N1 TX/24 HA in complex with the Fabs of 310-12D03, 310-1H02, 310-7D11, 326-289.74 and 326-366.26 produced good-resolution structures showing that each mAb engaged the HA1 head region with a distinct angle of approach and epitope location (Fig. 4a–c, Supplementary Figs. 1–5 and Supplementary Table 1).

While both 310-12D03 and 310-1H02 targeted the RBS, they employed distinct binding modes to engage common HA motifs. 310-12D03 engaged HA with all complementarity-determining

regions (CDR) of both heavy and light chains. Centred over the HA 130-loop, 310-12D03 also engaged residues from the HA 190-helix and 220-loop at the rim of the RBS, along with adjacent regions (Fig. 4d and Supplementary Table 2). Compared to 310-12D03, 310-1H02 targeted the HA RBS in a 90-degree rotated mode, engaging the HA 190-helix, 130-loop, 142-loop and 220-loop exclusively through heavy-chain-mediated interactions, with a smaller epitope of 614 Å$^2$ (Fig. 4e and Supplementary Table 2).

310-7D11 bound to the upper lateral region of the globular head, adjacent to the RBS. The heavy chain contributed two-thirds of the 900 Å$^2$ paratope surface with the 17-residue-long CDR H3, stabilized by a disulfide bond between C100 and C100$^E$, providing over 420 Å$^2$ binding surface. Interestingly, five of the six light-chain paratope residues (H27$^D$, S27$^E$, N28, Y30, Y32 and Y49) were directly encoded by the germline IGKV2D-28 gene, which explains the preferential usage of IGKV2D-28 in 310-7D11-like antibodies targeting the upper lateral epitope (Fig. 4f and Supplementary Table 2).

326-289.74 and 326-366.26 targeted the lower lateral region on HA, with epitopes partly overlapping that of 310-7D11 but shifted down towards the stem region (Fig. 4c). 326-366.26 bound to HA in a 90-degree rotated orientation relative to that of 326-289.74 and to a smaller epitope surface of 750 Å$^2$ (Fig. 4g,h and Supplementary Table 2). 326-289.74 and 326-366.26 recognized common HA motifs, such as the α1-β3 connecting loop, the β5 and β14, and the 140-loop, despite their distinct binding orientations (Fig. 4g,h and Supplementary Table 2).

## Vulnerability of the epitopes targeted by mAbs

We curated 523 non-redundant H5 HA amino acid sequences representing all the major clades from a total of 1,694 HA sequences of Gs/GD lineage H5Nx viruses to analyse the sequence conservation of the contact residues of the five H5 mAbs (Fig. 5a).

For both 326-289.74 and 326-366.26, E78$_{HA1}$, R80$^A_{HA1}$ and P82$_{HA1}$ comprise more than 1/3 of the total buried surface area (BSA) (Fig. 5a and Supplementary Table 2). While R80$^A_{HA1}$ in TX/24 HA is asparagine in many other H5 sequences, this substitution does not interfere with antibody binding or neutralization. Residues P123$_{HA1}$, K124$_{HA1}$ and S125$_{HA1}$ contributed to more than 1/3 of the total BSA, making 310-7D11 distinct from 326-289.74 and 326-366.26 (Fig. 5a). Variations in Vict/24 (2.3.2.1a) at residues Q121$^A_{HA1}$–R121$^A_{HA1}$ and/or S125$_{HA1}$–D125$_{HA1}$ might explain the inability of 310-7D11 to bind and neutralize this strain (Fig. 3b,e). Residues L133$_{HA1}$, V135$_{HA1}$, A137$_{HA1}$, G143$_{HA1}$ and P145$_{HA1}$ contributed more than 1/2 of the total BSA for both mAbs 310-12D03 and 310-1H02. While some of these positions exhibit a few variations, both RBS mAbs tolerate these variations. Both RBS mAbs also made contact with residues N224$_{HA1}$ and Q226$_{HA1}$. A Q226L mutation alone or in combination with N224K can switch receptor specificity from avian α2,3-linked to human α2-6-linked sialic acids[25], which could facilitate upper respiratory tract infection in humans. mAbs 310-12D03 and 310-1H02, however, still neutralized H5N1 TX/24 virus expressing HA with Q226L or N224K/Q226L mutations, indicating that they would maintain functionality if the virus were to acquire these mammalian adaptation mutations (Fig. 5b).

We also used a deep mutational scanning (DMS) non-replicative pseudovirus library based on HA of A/American wigeon/South Carolina/USDA-000345-001/2021 (clade 2.3.4.4b H5N1) to assess the neutralization resistance profile of every possible substitution at each HA position[26]. There was a high degree of agreement between the neutralization resistance maps and structurally defined epitopes (Fig. 6a), indicating that mutations outside the structurally defined epitopes are unlikely to cause neutralization resistance against the H5 mAbs. Of note, the neutralization resistance maps for these H5 mAbs are drastically different from that of animal sera after vaccination or infection with closely related H5N1 viruses[26], highlighting that the cross-clade neutralizing epitopes targeted by the H5 mAbs might not be easily targeted by primary vaccination or infection.

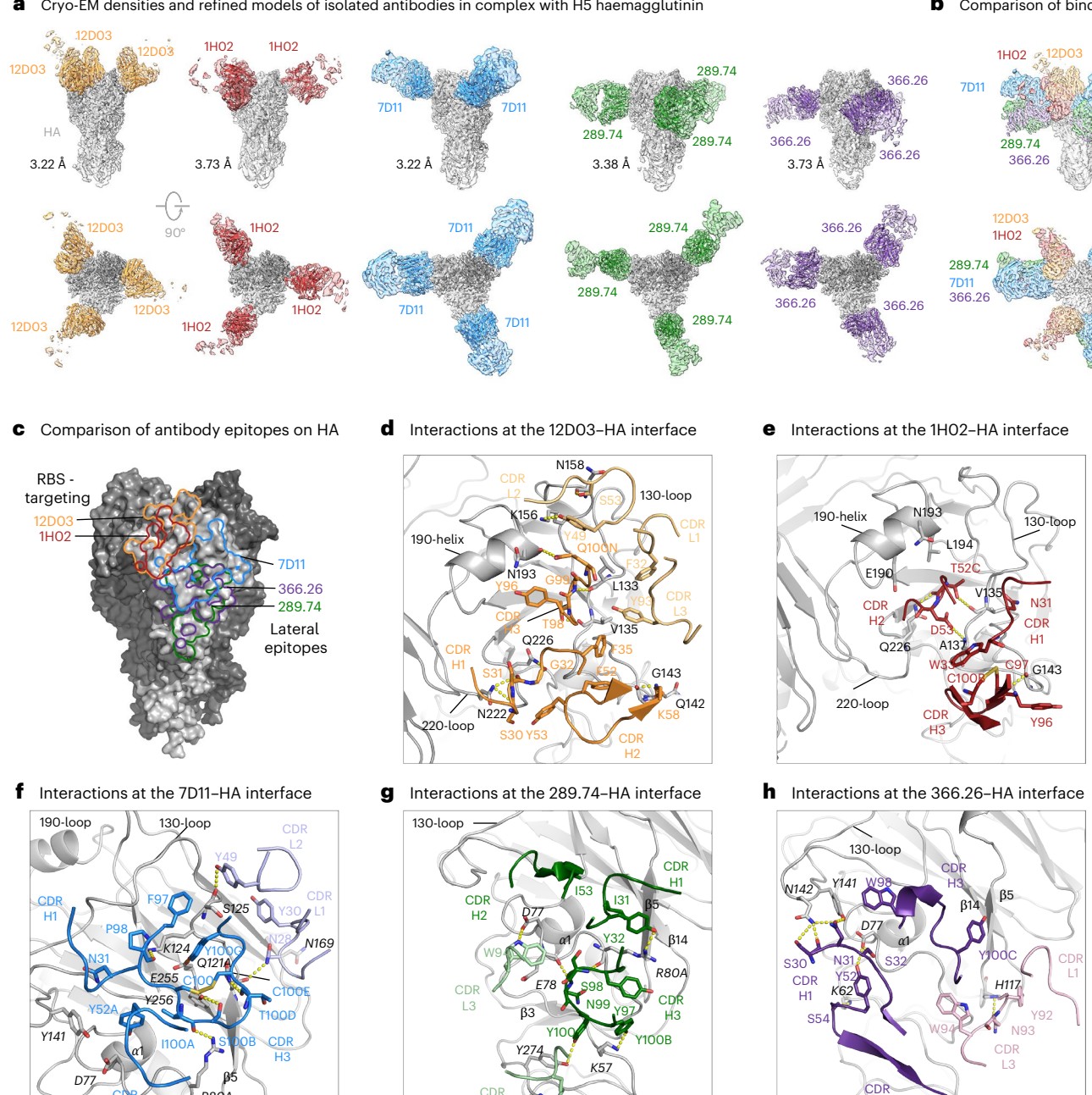

**Fig. 4 | Cryo-EM structures of H5N1 HA in complex with Fabs. a**, Cryo-EM densities and refined models. The refined models of the antigen-binding fragments of 310-12D03, 310-1H02, 310-7D11, 326-289.74 and 326-366.26 in complex with H5 HA are shown inside the semi-transparent electron density maps. Structures indicated that these antibodies targeted the RBS and the lateral epitope regions on HA. The protomers of the trimeric HA are shown in different shades of grey, and the Fabs are coloured in orange, firebrick red, dodger blue, green and purple for 310-12D03, 310-1H02, 310-7D11, 326-289.74 and 326-366.26, respectively. Only the variable regions of the Fabs were built and refined. **b**, Superposition of the cryo-EM density maps. The maps were superposed on the HA regions and coloured in the same scheme as in **a. c**, Epitopes of antibodies 310-12D03, 310-1H02, 310-7D11, 326-289.74 and 326-366.26 on HA. The footprints

of each antibody are marked with lines coloured in the same scheme as in **a**. Both 310-12D03 and 310-1H02 bound to the RBS on HA, whereas 326-289.74 and 326-366.26 targeted the lateral region below the RBS. 310-7D11 bound to an area overlapping with the RBS and lateral epitope regions. **d**–**h**, Interactions at the antibody–HA interfaces. The antibody elements contributing to the binding are shown as cartoons and coloured in the same colour scheme as in **a**. The light-chain elements are shown in lighter shades than corresponding heavy chains for clarity. Residues with substantial contributions to the binding interactions are shown in stick representation, with identified hydrogen bonds highlighted in yellow dashed lines. Key HA structural motifs, including the 190-helix, 130-loop and 220-loop, which contribute to the formation of the HA RBS, are highlighted for positional reference.

To further investigate the potential of the 2.3.4.4b H5N1 virus to escape from the H5 mAbs, we serially passaged our rewired replication-restricted reporter influenza virus[27] expressing TX/24 HA and neuraminidase without mAb pressure or with increasing concentrations of 326-289.74, 310-7D11 or 310-12D03, followed by analysis of virus

sequence variants using high-throughput single-genome sequencing (HT-SGS)[28,29]. This engineered virus replicates only when the viral polymerase (PB1) is complemented in *trans* by infected cells and does not facilitate genomic RNA reassortment of the HA-encoded segment due to genomic packaging signal incompatibility[27,30], enabling us to safely

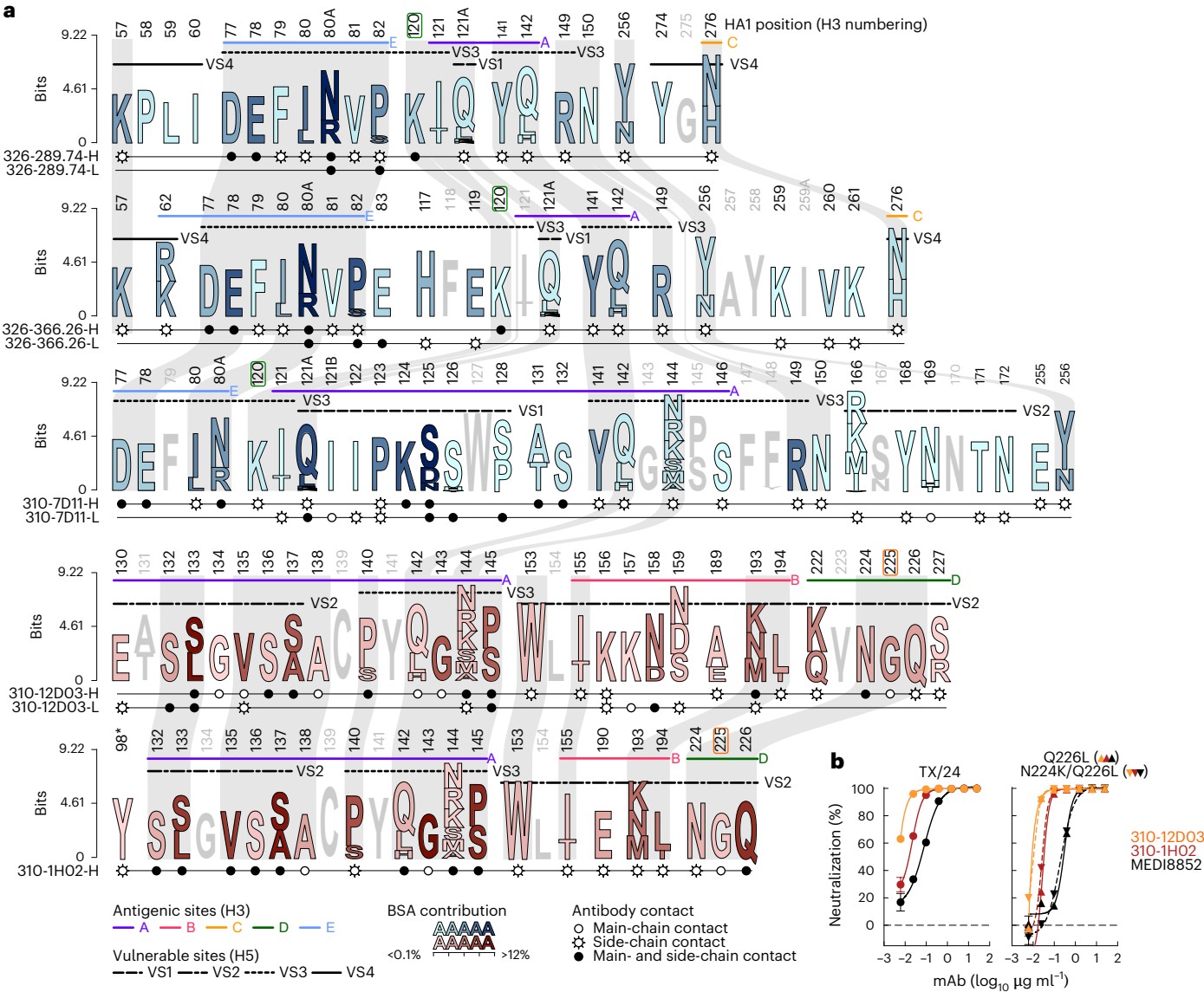

**Fig. 5 | Sequence conservation and mutation tolerability of the epitopes targeted by top mAbs. a**, Amino acid sequence variations of the epitopes targeted by top mAbs. HA residues that have contact with each of the five mAbs are shown in the sequence logos. The sequence logo was generated with a collection of 523 non-redundant gsGD H5Nx HA sequences. Each residue is coloured according to the buried surface area at the HA–Fab interface. Types of interaction between HA and Fab residues are indicated with symbols below each residue. Shared epitope residues between mAbs are shaded and connected. HA antigenic sites based on H3 (ref. 41) as well as vulnerable sites (VS)[36] are indicated above the sequence logos. **b**, Neutralization sensitivity of TX/24 viruses carrying sialic acid receptor preference switching mutations to RBS mAbs. Anti-HA stem MEDI8852 was used as a control. Triangle symbols with solid lines and reverse triangles with dashed lines indicate neutralization curves for Q226L and N224K/Q226L viruses, respectively. Data represent the mean ± s.d. of 4 technical replicates, representative of 2 independent experiments.

generate antibody escape viral variants. Selection with mAb 326-289.74 was associated with emergence of sequence variants bearing a K120T$_{HA1}$ substitution beginning at round 3 (R3) (Fig. 6b, Extended Data Fig. 7 and Supplementary Table 3). After 5 passages (R5), the selected virus exhibited neutralization resistance not only against 326-288.74 but also against 326-366.26, and a ≥100-fold reduced neutralization sensitivity to 310-7D11 (Fig. 6c), although K120$_{HA1}$ minimally contributed to the 310-7D11 footprint (Figs. 5a and 6a). In contrast, virus grown in the presence of 310-7D11 did not acquire neutralization resistance to this mAb in two independent attempts. Selection with mAb 310-12D03 was associated with emergence of variants bearing G225$_{HA1}$ to glutamic acid from a haplotype with a non-synonymous mutation (Fig. 6d, Extended Data Fig. 7 and Supplementary Table 3). This selected virus exhibited complete neutralization resistance against 310-12D03, but neutralization sensitivity to the other RBS mAb 310-1H02 remained

unchanged (Fig. 6e). Importantly, neither 326-289.74-selected nor 310-12D03-selected virus acquired a replication fitness advantage over the unselected TX/24 virus when passaged with the parental virus in the absence of mAb (Fig. 6f and Supplementary Table 4). Taken together, while virus escape from lower-lateral-targeting mAbs and RBS-targeted 310-12D03 may be more likely, viral escape against the potent mAb 310-7D11 targeting the upper lateral region appears to be more difficult.

## mAb cocktails confer protection from lethal H5N1 challenge
Because 310-12D03 selected for escape mutations in HA, we tested whether combinations of mAbs could both protect in vivo and mitigate viral escape. After 10 rounds of selection in the presence of both 310-7D11 and 310-12D03, we were unable to generate neutralization resistant TX/24 virus, suggesting that simultaneously escaping from both these mAbs is difficult. We next tested whether combining mAbs

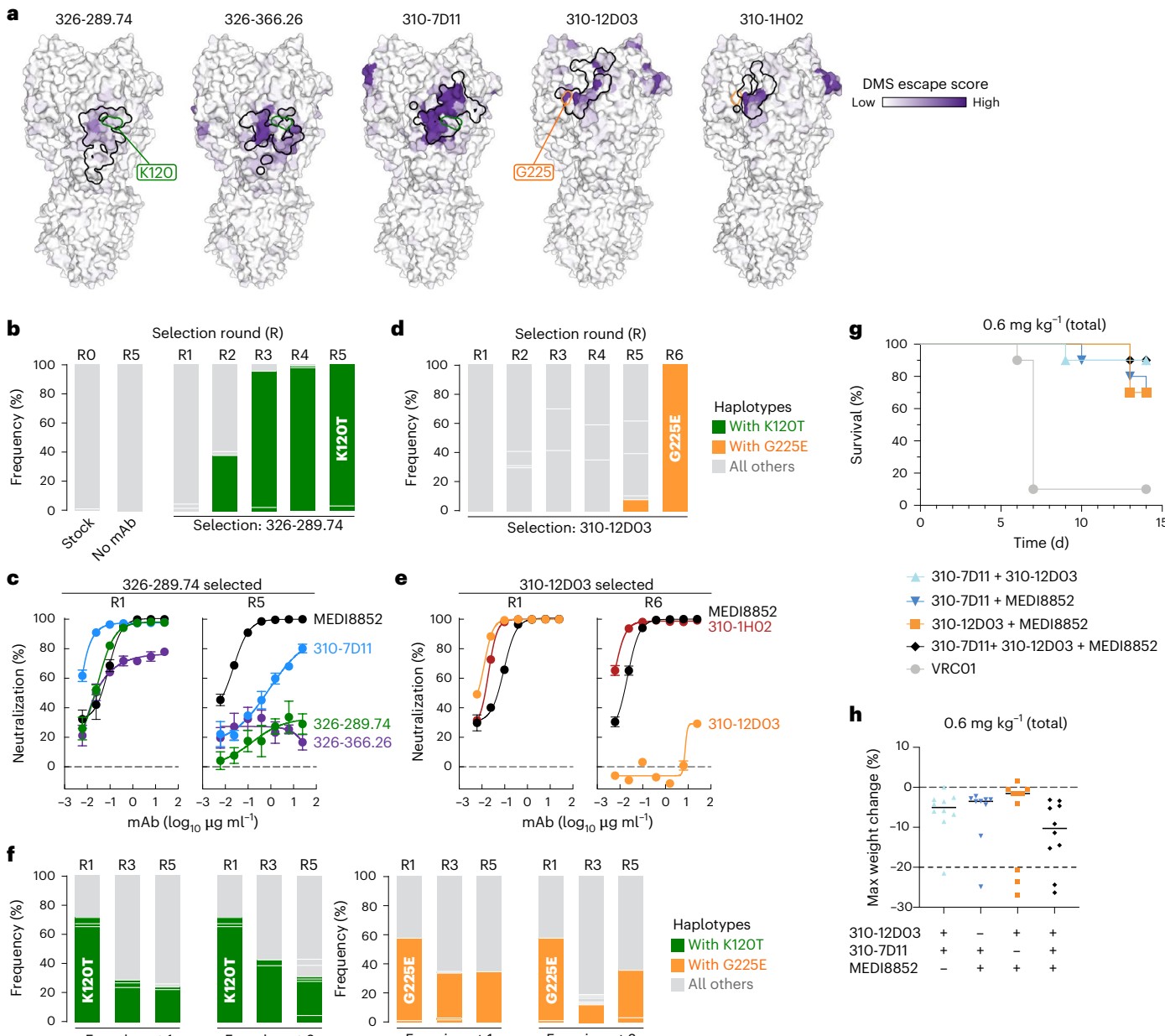

**Fig. 6 | Virus escape generation and in vivo protection with co-administered mAbs. a**, DMS library-based selection of mAb escape. Lentivirus-based pseudovirus expressing H5 DMS HA library was used to enrich mAb escape variants. DMS escape score is shown in purple gradient. The structurally defined epitope of each antibody is shown as a black outline. See the interactive version of deep mutational scanning data at https://dms-vep.org/Flu_H5_American-Wigeon_South-Carolina_2021-H5N1_DMS/htmls/VRC_antibody_escape_faceted.html. The residues K120 and G225 are highlighted in green and orange outlines, respectively. **b**, HA haplotype frequency of TX/24 virus in the absence or presence of 326-289.74 pressure by single-genome sequencing. Samples from input TX/24 virus stock as well as virus culture without mAb (no mAb) were used as controls. In each successive round, an increased amount of selection mAb was used. **c**, Neutralization sensitivity of TX/24 viruses selected with 326-289.74. Neutralization curves of early (R1) and late (R5) TX/24 viruses against mAbs. MEDI8852 was used as a control. **d**, HA haplotype frequency of TX/24 virus in the absence or presence of 310-12D03 pressure by single-genome sequencing.

**e**, Neutralization sensitivity of TX/24 viruses selected with 310-12D03. Neutralization curves of early (R1) and late (R6) TX/24 viruses against mAbs. **f**, Viral replication fitness of TX/24 and its neutralization resistant variant carrying either K120T or G225E mutation. HA haplotype frequency after 1:1 mix of TX/24 virus and either 326-289.74-selected virus (left) or 310-12D03-selected virus (right) (R1) and after serial passages in the absence of antibody pressure. R3 and R5 samples were taken from 2 independent replicates after 3 and 5 passages, respectively. **g**, Protection against H5 TX/24 challenge of combinations of mAbs. Total mAb dose including all mAbs in each condition was 0.6 mg kg$^{-1}$. Statistical significance of survival was determined by log-rank (Mantel–Cox) test with Bonferroni–Sidak adjustment. All mAb combinations were statistically significant relative to VRC01 ($P < 0.001$); $P = 0.0002$ for 310-7D11 + 310-12D03, $P = 0.0006$ for 310-12D03 + MEDI8852, $P = 0.0008$ for 310-7D11 + MEDI8852 and $P = 0.0002$ for 310-7D11 + 310-12D03 + MEDI8852. **h**, Maximum weight change (%) post challenge with same mAb combinations as in **g** as indicated. There was no significant difference between groups by Kruskal–Wallis test.

binding distinct epitopes could protect against H5 TX/24 challenge. We administered 0.3 mg kg$^{-1}$ of 310-12D03 together with 310-7D11, or each of these mAbs with MEDI8852. We also administered 0.2 mg kg$^{-1}$ of all three mAbs so that a total of 0.6 mg kg$^{-1}$ mAb was given in each

case. 310-12D03 and 310-7D11 together conferred 90% protection with or without further addition of MEDI8852, while combinations of one head mAb and MEDI8852 provided 70% protection (Fig. 6g,h and Extended Data Fig. 8). There was no statistical difference in

survival among combination groups, but each combination, except 310-12D03 + MEDI8852, was more protective than with each antibody alone at 0.3 mg kg$^{-1}$ (Fig. 6g,h and Extended Data Fig. 8). Each antibody at 0.6 mg kg$^{-1}$ alone conferred similar protection as mAb cocktails, suggesting an additive rather than a synergistic effect (Fig. 3g,h and Extended Data Fig. 8). We conclude that combinations of mAbs binding distinct epitopes could provide strong protection to infection in vivo and limit viral escape.

## Discussion

Should HPAI clade 2.3.4.4b H5N1 viruses acquire the necessary mammalian adaptations to facilitate human-to-human transmission, we will likely have a global health emergency. Pre-existing immunity from seasonal and pandemic H1N1 influenza exposure could attenuate disease severity[31], but given the ~50% morbidity experienced in sporadic avian-origin human H5N1 infections in Southeast Asia, seasonal influenza exposure cannot fully mitigate serious human disease with HPAI H5N1 (ref. 8). Vaccines derived from H5Nx clade 2.3.4.4b candidate vaccine viruses (CVVs) are authorized for use in Canada[32] and Europe[33], and early data suggest that CVV A/Astrakhan/2020 H5N8-based vaccines induce seroprotective titres in humans and can provide cross-protection against currently circulating 2.3.4.4b H5N1 (ref. 34). In the advent of a pandemic, however, early intervention apart from vaccines, such as antibody biologics, will be crucial in limiting morbidity and mortality induced by an HPAI virus, especially in high-risk populations.

Before this study, several mAbs targeting the H5 HA head had been characterized, but most are unlikely to bind or potently neutralize H5 TX/24 (refs. 35–38). Recently, a panel of H5 antibodies was generated from humanized mice vaccinated with 2.3.4.4b H5 HA and N1 NA that bind H5 TX/24 but with limited breadth to other H5 clades[39]. While the antibodies neutralized and protected against a 2022 avian 2.3.4.4b HPAI H5N1 strain, neutralization of bovine H5 TX/24 was not shown[39]. Here we undertook isolation and characterization of mAbs expressed by B cells elicited by vaccination in a human clinical study with H5 Indo/05 that also cross-reacted with H5 TX/24, with the aim of finding the most potent and broad H5-head-specific mAbs. Despite the antigenic mismatch between the vaccinating H5N1 strain driving affinity maturation and H5 TX/24, antibodies isolated at 6 months p.b. after vaccination had higher neutralization potency to H5 TX/24 compared with antibodies isolated at 2 weeks p.b. Five mAbs displayed marked breadth among diverse clades of H5Nx human viral isolates, including representatives of clade 2.3.2.1e viruses circulating in Southeast Asia and panzootic clade 2.3.4.4b viruses circulating in Europe and the Americas. Although MEDI8852 is considered one of the best HA stem-directed mAbs isolated so far[20], it was 20-fold less potent compared to these mAbs. Mice that received 310-12D03 and 310-7D11 prophylactically had similar survival rates as MEDI8852 after clade 2.3.4.4b H5N1 infection, but we noted substantially more weight loss in animals that received MEDI8852, suggesting more severe disease in these animals. This is probably due to differences in mechanisms of neutralization. While 310-12D03 and 310-7D11 block sialic acid binding to the RBS and viral entry, HA stem-directed mAbs primarily inhibit membrane fusion after viral attachment and endocytosis, allowing innate immune sensing and inflammatory responses to occur. Further studies including measurement of lung titres are needed to fully understand the mechanism and extent of protection for HA head and stem mAbs in this highly pathogenic H5N1 challenge mouse model. We also do not address the ability of the mAbs to protect against viral transmission.

One concern with HA head-targeted mAbs is viral escape, given the high mutability of many HA head epitopes. Viral escape studies demonstrated that H5N1 TX/24 virus could escape from 326-289.74 and 310-12D03 in vitro, but we could not detect escape from 310-7D11 either alone or as a cocktail with 310-12D03. These experiments and animal

protection studies with multiple antibody cocktails binding distinct epitopes suggest that these mAbs in combination or in a bispecific format could provide broad and potent protection across H5 clades and be resistant to antibody-mediated viral escape.

We note that the lead mAbs identified in this study did not target classical antigenic sites characterized in the literature for H1 (ref. 40), H3 (ref. 41) or H5 (ref. 36), or epitopes targeted by serum antibodies through primary immunization or infection with closely related clade 2.3.4.4b H5N1 viruses in mice or ferrets[26]. We specifically selected for mAbs cross-reactive for both Indo/05 and TX/24, which allowed for the identification of highly potent and cross-neutralizing H5 mAbs such as 310-12D03 or 310-7D11. However, such mAbs may not be efficiently elicited through primary H5N1 immunization or infection. It may require targeted elicitation of antibodies to the RBS and/or upper lateral epitopes through structural vaccine design such as epitope cloaking[42], epitope masking[42] and diversified mosaic antigen display[43–45]. Our mAbs were markedly more potent at 6 months p.b., but they had relatively low percent SHM and affinity. The low affinity and limited epitope footprint of the RBS Fabs might be desired to circumvent hypervariability surrounding the RBS, as it might allow for tolerance of variations as described for other RBS mAbs[46,47]. Nevertheless, efforts to affinity mature those antibodies by directed evolution or protein design may yield higher affinity and potentially more potent and/or broader neutralizing variants.

In summary, we conducted a highly effective human antibody discovery campaign that yielded several lead mAb candidates with high neutralizing potency and broad cross-clade breadth against H5Nx viruses, including the clade 2.3.4.4b virus. The lead mAbs identified in this study offer direct translational value for H5N1 influenza pandemic preparedness alongside ongoing vaccine development efforts.

## Methods

### Ethics statement

All clinical trials were reviewed and approved by the National Institutes of Health institutional review board. For pharmacokinetic studies, the study protocol was reviewed and approved by the Institutional Animal Care and Use Committee (IACUC) of the Vaccine Research Center (VRC), National Institute of Allergy and Infectious Diseases, National Institutes of Health. All mice in this study were housed in Association for Assessment and Accreditation of Laboratory Animal Care (AAALAC)-accredited animal facilities in a 12-h light/dark cycle at an ambient temperature of 22.2 ± 2.8 °C with a relative humidity maintained between 30 and 70%. For passive transfer and viral challenge studies, animal procedures were approved by the University of Pittsburgh IACUC, with animal care in accordance with the Guide for the Care and Use of Laboratory Animals (National Research Council) and AAALAC, and studies were conducted in a biosafety level 3 facility.

### Vaccine study design

The H5 vaccine study (NCT01086657)[14] and the FluMos-v2 study (NCT05968989) were conducted at the NIH Clinical Center by the VRC Clinical Trials Program of NIAID. Written informed consent was obtained from every enrolled individual and complied with all relevant ethics regulations. Consent included sample use for broad purposes in future research, including the potential development of new products. Compensation was given for time and effort related to participation in the clinical trial. Both studies were phase 1, open-label, randomized clinical trials in healthy adults designed to study the safety, tolerability and immunogenicity of prime–boost vaccination regimens. In the H5 vaccine study, individuals were vaccinated with a recombinant DNA plasmid that encodes H5 A/Indonesia/05/2005 HA or a monovalent influenza subunit virion (MIV; A/Indonesia/05/2005) vaccine manufactured by Sanofi Pasteur. All individuals were then boosted 4–24 weeks later with the MIV vaccine. In the FluMos-v2 vaccine study, individuals were vaccinated with a mosaic nanoparticle displaying 20

HA ectodomain trimers from the following influenza strains: H1 A/Idaho/07/2018, H2 A/Singapore/1/1957, H3 A/Perth/1008/2019, H3 A/Darwin/106/2020, B/Victoria B/Colorado/06/2017 and a B/Yamagata B/Phuket/3073/2013 strain. All individuals were then boosted 16 weeks later with the mosaic nanoparticle. Only one individual who was a participant of the FluMos-v2 trial was included in this study.

## Protein expression and purification
mAbs and recombinant HA proteins were expressed in Expi293F (Thermo Fisher, A14527) cells by transient transfection using Expifectamine transfection reagents (Thermo Fisher). HA antigens contained a point mutation at the sialic acid-binding site (Y98F) within the HA ectodomain, a T4 fibritin foldon trimerization domain, a hexahistidine tag and an AVI-tag. mAb heavy and light-chain sequences were synthesized and cloned by GenScript into IgG1 kappa or lambda bicistronic expression vectors. For the pharmacokinetics study, LS mutations (M428L/N4324S) were introduced in the IgG1 Fc domain[48]. Cultures were suspended for 4–7 days at 37 °C and 8% $CO_2$ saturation with shaking. For HAs, at collection, cultures were centrifuged at 2,500 × $g$ for 20 min at 4 °C, followed by filtration (0.45 µm) and then slow purification by metal affinity chromatography. For each litre of supernatant, 10 ml of Ni Sepharose Excel resin (GE) bed was washed (10× resin bed volume) with buffer composed of Tris (50 mM, pH 8.0), NaCl (500 mM) and imidazole (30 mM, pH 8.0) using a gravity column, followed by slow dripping of clarified supernatant at 4 °C. A final column wash (>10× bed volume) with Tris (50 mM, pH 8.0), NaCl (500 mM) and imidazole (30 mM, pH 8.0) preceded elution of His-tagged protein using 50 mM Tris (pH 8.0), 500 mM NaCl and 300 mM imidazole. Eluates were concentrated and applied to a HiLoad 16/600 Superdex 200 pg column pre-equilibrated with PBS + 0.01% sodium azide for preparative size exclusion chromatography. Fractions containing trimeric HA were identified on the basis of elution volume and verified with SDS–PAGE. Fractions were pooled, concentrated and stored at −80 °C until use.

Monoclonal antibodies were purified from the cell supernatant using sepharose Protein A (Pierce). To produce Fabs for kinetics, the mAbs were cleaved by LysC enzyme (1:2,000 w/w) (New England Bio-Labs) at room temperature overnight. The enzymatic digestion was stopped using protease inhibitor (Roche). The digestion mixture was then passed through a protein A or protein G column to separate the Fc fragment from the Fab.

## Biotinylation and labelling of antigens
Antigens were dialysed into 10 mM of Tris-HCl buffer using Thermo Scientific Slide-A-Lyzer mini dialysis devices (3.6 KDa molecular weight cut-off (MWCO), 88400) before biotinylation using BirA enzyme–biotin ligase and reaction buffers (Avidity, BirA500). Final reaction mixtures contained 1× BioMixA, 1× BioMixB, AVI-tagged protein and BirA enzyme. Reactions were incubated for 1 h with mixing at 600 r.p.m. at 30 °C before removal of excess biotin with Amicon (30 KDa MWCO) columns. Biotinylated antigens were conjugated to streptavidin-labelled fluorochromes for detection of HA-specific B cells.

## Flow cytometry and single-cell sorting for RATP-Ig
Cryopreserved PBMCs from blood collected at 2 weeks and 6 months p.b. from trial participants were stained with a variety of cell markers and HA probes for fluorescence-activated cell sorting (FACS). Cell markers included: CD3 (1:400 dilution), CD56 (1:200 dilution), CD14 (1:200 dilution) and CD20 (1:400 dilution) from BioLegend; IgG (1:100 dilution) and IgM (1:40 dilution) from BD Biosciences; and CD19 (1:50 dilution) from Beckman Coulter and Aqua dead for live/dead discrimination (Thermo Fisher). Soluble fluorochrome-labelled HAs included H5 A/Indonesia/05/2005 ectodomain and stabilized stem; H5 A/Texas/37/2024 ectodomain; H1 A/New Caledonia/20/1999 ectodomain; and H1 A/California/04/2009 ectodomain. Stained samples were

run on a FACSAria II (BD Biosciences) running BD FACSDiva software 8.0, and data were analysed using FlowJo v.10 (TreeStar). CD3⁻CD14⁻CD56⁻CD19⁺CD20⁺IgG⁺IgA⁺IgM⁻ memory B cells were gated for isolation of HA-binding memory B cells. Cells were single-cell sorted into 96-well plates for RATP-Ig processing[15]. Briefly, single-cell RNA was purified for complementary (c)DNA synthesis, followed by immunoglobulin enrichment. Enriched products are sequenced on an Illumina NextSeq 2000 and assembled into linear DNA expression cassettes including a CMV promoter, heavy chain or light chain constant region and polyA tail. Amplified DNA cassettes were transfected into Expi293 cells in 96-well deep-well plates using the Expi293 Expression System according to manufacturer protocol. Cultures were incubated with shaking at 1,100 r.p.m. for 5–7 days at 37 °C and 8% $CO_2$. Supernatants were clarified by centrifugation and then collected.

## V(D)J sequence analysis
Reads were demultiplexed using bcl2fastq v.2.20.0.422 and analysed for V(D)J sequences as previously described[15], using SONAR v.4.3 to annotate and cluster the sequences. Paired heavy and light-chain sequences were clustered into groups based on 80% identity.

## Phylogenetic analysis of HAs used for binding assays
HA amino acid sequences from HPAI H5Nx and H2N2 and H1N1 matching the recombinant HA proteins used for binding assays (11 in total) were downloaded from NCBI and the GISAID EpiFlu database. Multiple sequence alignments were performed using ClustalW, and the maximum-likelihood tree was inferred using a JTT matrix-based model with a discrete Gamma distribution to model evolutionary rate differences among sites. Evolutionary analyses were conducted in MEGA11 (ref. [49]).

## H5 phylogenomic tree analysis
HA sequences were downloaded from NCBI and the GISAID EpiFlu database. A total of 1,694 Gs/GD H5 HA sequences were filtered with HHfilter using 99% cut-off value to curate 523 non-redundant sequences[50,51]. Sequence logo plots were generated using Skylign tool[52].

## Meso Scale Discovery (MSD)
MSD multi-array 384-well streptavidin-coated SECTOR Imager 600 plates were coated with 5% MSD Blocker A, then incubated for 1 h at room temperature before being washed with wash buffer (PBS + 0.05% Tween). Plates were coated with biotinylated HA protein diluted to 1 µg ml⁻¹ in 1% MSD Blocker A and incubated for 1 h, then washed. Antibodies were diluted in 1% MSD Blocker A before being added to the plate and incubated for 1 h. RATP-Ig supernatants were diluted 1:10, and mAbs were diluted to a concentration of 1 µg ml⁻¹ before being serially diluted threefold. Plates were then washed, coated with SULFO-TAG goat anti-human IgG at a concentration of 2 µg ml⁻¹, and incubated for 1 h before washing. Plates were coated with 1× MSD reading buffer and read on an MSD SECTOR S 600MM imager. For the serially diluted mAbs, binding curves were plotted using GraphPad Prism 10 and area under the curve was calculated. HA from the following strains were tested: H5N1 A/Texas/37/2024, H5N1 A/Indonesia/02/2005, H5N1 A/Cambodia/2302009/2023, H5N1 A/Vietnam/1194/2004, H5N6 A/Fujian/2/2024, H5N6 A/Sichuan/26221/2014, H5N1 A/Victoria/149/2024, H1N1 A/New Caledonia/20/1999, H1N1 A/California/04/2009, H2N2 A/Singapore/1/1957 and H2N2 A/Ann Arbor/7/1967.

## Competitive binding of H5 mAbs with site-specific antibodies
We coated 96-well plates with 100 µl of 2 µg ml⁻¹ purified H5 TX/24 ectodomain at 4 °C overnight. Plates were then blocked with 200 µl of 1% BSA in PBS + 0.05% Tween (PBS-T) for 1 h at room temperature and washed three times with PBS-T. Preparations of select Fabs of interest were added to wells at a final concentration of 10 µg ml⁻¹ (75 µl per well) and incubated for 1 h at room temperature. Recombinantly expressed

IgGs were then added to wells of each Fab at final concentrations of 0.25 µg ml$^{-1}$, 0.025 µg ml$^{-1}$ and 0.0025 µg ml$^{-1}$ in a volume of 25 µl per well, without washing the competitor Fab, and then incubated for 1 h at room temperature. Plates were washed three times with PBS-T and bound antibodies were detected using horseradish peroxidase (HRP)-conjugated goat anti-human IgG Fc (1:15,000, Invitrogen) and 1-Step TMB ELISA substrate (Thermo Scientific).

## Reporter viruses

H5Nx viruses (that is, H5N1 A/Vietnam/1203/2004, A/Indonesia/05/2005, A/Cambodia/2302009/2023, A/Victoria/149/2024; A/Texas/37/2024; H5N6 A/Sichuan/26221/2014; and non-GsGD H5N2 A/State of Mexico/INER-INF645/2024) and H2N2 A/Singapore/1/1957 virus used for the neutralization assay, as well as H5N1 A/Texas/37/2024 used to generate virus escape mutants were prepared as rewired replication-restricted reporter viruses (R4ΔPB1). The procedure for rescue and propagation of these viruses is described in detail elsewhere[27,29,30]. Briefly, R4ΔPB1 H5Nx viruses had the PB1 segment engineered by inserting the coding region of the H5 (or H2) HA gene between the PB1 genomic packaging signals, thereby missing a functional PB1 gene. Inserted H5 (or H2) HA sequences were modified by introducing synonymous mutations to inactivate the HA genomic packaging signals, which comprise the 16 amino acids of the signal peptide and the last 12 amino acids of the cytoplasmic domain, and the polybasic cleavage site was replaced with a monobasic cleavage site to enable virus propagation only in the presence of exogenous trypsin[53]. The HA segment was prepared by inserting the fluorescent reporter TdKatushka2 gene fused with a nuclear localization signal at the C terminus between the sequences containing the HA genomic packaging signals of A/Puerto Rico/8/1934 (Genbank MW298214.1). The internal genes of A/WSN/1933 H1N1 influenza virus (that is, PB2, PA, NP, M and NS) were used to rescue these viruses. R4ΔPB1 viruses were only able to propagate in MDCK-SIAT1 (Millipore Sigma, 05071502) cells constitutively expressing PB1 of A/WSN/1933 (MDCK-SIAT-PB1 cell line) in the presence of 1 µg ml$^{-1}$ TPCK-treated trypsin (Sigma). Due to genomic packaging signal incompatibilities, R4ΔPB1 viruses used in the study do not generate replication-competent virus through reassortment of viral genomic RNA segments with wild-type influenza viruses. These two layers of safety precaution (missing PB1 gene and rewired segments) ensure compliance to biosafety standards. R4ΔPB1 viruses were titrated to determine the linear range between the virus dilution and the number of infected cells[54].

## Neutralization

Neutralization activity of antibodies was determined using R4ΔPB1 H5Nx viruses as previously described[27,54]. Briefly, either 2 dilutions (for screening RATP-Ig supernatants) or 7 fourfold serial dilutions of antibody were mixed with an equal volume of pre-titrated R4ΔPB1 virus in 96-well U-bottom untreated plates. Of note, virus titrations and neutralization assays were performed without adding exogenous trypsin. After 1 h incubation at 37 °C in 5% CO$_2$ humidified atmosphere, 30 µl MDCK-SIAT1 cells (Millipore Sigma, 05071502) expressing PB1 (8 × 10$^5$ cells per ml) cells were added to 90 µl antibody–virus solution, and the mixture of antibody–virus–cells was transferred to a 384-well plate in quadruplicate (25 µl per well). The 384-well plates were then incubated overnight at 37 °C in 5% CO$_2$ humidified atmosphere, and fluorescent cells were imaged and counted using a Celigo Image cytometer (Revvity) with a customized red filter for detecting TdKatushka2 fluorescence[27]. Percent neutralization was calculated for each well by constraining the virus control (virus plus cells) as 0% neutralization and the cell-only control (no virus) as 100% neutralization. A 7-point neutralization curve was plotted against antibody concentration for each sample, and a sigmoidal 4PL curve fit generated using Prism was used to estimate the 80% inhibitory concentrations. These concentrations take into account the antibody dilution after admixing with virus at a 1:1 ratio.

## Viral escape

Selection and enrichment of virus escape mutants towards neutralizing antibodies utilized the R4ΔPB1 H5 TX/24 reporter virus. Serial dilutions of antibody were mixed with an equal volume of pre-titrated virus. After incubation, pre-washed MDCK-SIAT1-PB1 cells with or without addition of TPCK-treated trypsin were added to the antibody–virus mixtures and transferred to a 96-well plate. For each antibody dilution set, one set contained TPCK-treated trypsin (1 µg ml$^{-1}$ final concentration) and one set did not, giving a set whereby virus undergoes a single cycle of replication and a set where virus can propagate for multiple cycles of replication. Plates were imaged at 24-h increments as described above. Well pairs for a given antibody concentration that showed >90% inhibition in the single-cycle condition and >20,000 fluorescent cell counts in the propagation condition were chosen to collect virus for the next round. Once the antibody being selected against exhibited an IC$_{50}$ greater than 25 µg ml$^{-1}$, virus was sequenced.

## Deep mutational scanning

Pseudovirus-based deep mutational scanning libraries for H5 haemagglutinin using A/American Wigeon/South Carolina/USDA-000345-001/2021 strain have been described previously[26]. This deep mutational scanning uses libraries of HA expressed on pseudotyped lentiviral particles that can only undergo a single round of infection, and can thus be used to safely study HA mutations at biosafety level 2. To identify mutations that lead to escape, each antibody was incubated with the deep mutational scanning library at concentrations 3 times and 12 times above their IC$_{99}$ value (as determined by a pseudovirus neutralization assay) for 45 min at 37 °C. After incubation, virus library–antibody mixtures were used to infect 293T cells (ATCC), viral genomes were recovered at 15 h post infection and DNA libraries were prepared for Illumina sequencing. Two biological replicates (virus libraries with independent mutation sets) were used for each antibody escape mapping.

To calculate escape caused by each variant in the library, a non-neutralizable control was used as described previously[55]. Escape for individual mutations in the library was calculated using a biophysical model implemented in the 'polyclonal' package (https://jbloomlab.github.io/polyclonal/)[56], as in our previous work with this library[26].

## Haemagglutinin inhibition assay

mAbs were diluted in a 96-well format with PBS (total final volume of 25 µl per dilution) and 25 µl of pre-titrated H5N1 R4ΔPB1 viruses (at a concentration of 8 HA units per 50 µl) was added to each well with mixing. After a 30-min incubation at room temperature, 50 µl of 0.5% pre-washed turkey red blood cells (Lampire Biological) was added to each well. The samples were allowed to incubate for an additional 30 min at room temperature, and the mAb dilution that no longer completely inhibited haemagglutination as seen by drip test was documented as the HAI titre for each sample.

## Sample extraction and HT-SGS

Viral samples were extracted and inactivated using the Quick-RNA MagBead (Zymo, R2132) protocol, substituting beads for MagBead J beads (Zymo, D4100-3-3). The protocol was modified to enhance recovery of genetic material from smaller sample volumes and lower RNA concentrations, using 25% of suggested volumes. Each sample used 10 µl of MagBead J. Both the stock and the passage without antibody exposure were also sequenced to distinguish natural or stock-based adaptation from antibody-driven escape.

HT-SGS of the HA segment was performed following modified protocols described previously[28,29]. Primer and probe adaptations are listed in Supplementary Table 5. The extension temperature for cDNA synthesis was set to 58 °C, and 20,000 cDNA copies were targeted per sample for PCR. Amplicons were prepared and barcoded for sequencing using the SMRTbell prep kit v.3.0 (PacBio, 102-182-700) and SMRT-bell Barcoded Adapter plate 3 (PacBio, 102-009-200) according to the

protocol (PacBio, 102-359-000 REV04 DEC 2024), followed by the Revio polymerase kit (PacBio, 102-817-600). The libraries were sequenced on the PacBio Revio.

## Sequence data processing

PacBio circular consensus sequences (CCS) generated with SmrtLink v.13 were mapped against the reference HA sequence A/Texas/37/2024 to identify the HA segment using Perl v.5.16.3, RStudio v.2022.07.1 and Minimap2. Then, single genome sequences (SGS) were generated following the UMI PacBio pipeline (https://github.com/niaid/UMI-pacbio-pipeline). SGS sequences were further cleaned and haplotypes with a minimum of 5× SGS coverage or a 0.1% SGS frequency were called using Perl v.5.16.3, SeqKit (v.2.3.1)[57], Cutadapt v.4.0, MAFFT (v.7.467)[58] and Geneious Prime v.2023.0.4 (Biomatters). Statistics and visualization were performed with RStudio v.2022.07.1. A phylogenetic tree was generated from all unique HA sequences using the maximum-likelihood method, employing the program RAxML[59]. The substitution model used was GTR + G + I with invariable sites, and bootstrap support was generated with rapid bootstraps using 1,000 replications. The resulting phylogeny was visualized with ITOL[60].

## Surface plasmon resonance (SPR)

High-throughput SPR capture kinetic experiments were performed on a Carterra LSA system, using an SAHC30M sensor chip in a 384-ligand array format. The chip surface was primed with run buffer (HBST-Carterra) and conditioned using 50 mM NaOH (Carterra), 1 M NaCl (Teknova) and 10 mM glycine pH 2 (Carterra). A capture lawn was prepared using 15 µg ml$^{-1}$ of Human Fab-kappa Kinetics Biotin and Human Fab-lambda Kinetics Biotin conjugate mix (Thermo Scientific) for 10 min, followed by a 10 mM glycine pH 2 wash to determine capture lawn viability. For capture kinetics, Fabs were diluted fourfold (200 nM, 50 nM, 12.5 nM, 3.125 nM) in HBST + 0.5 mg ml$^{-1}$ BSA (HBST + BSA) running buffer and printed onto the capture lawn for 7 min, followed by a baseline injection of running buffer (HBST + BSA). Antigens of interest (H5 A/Texas/37/2024, H5 A/Sichuan/26221/2014, H5 A/Victoria/149/2024, H5 A/Vietnam/1194/2004, H5 A/Cambodia/2302009/2023 and H5 A/Indonesia/05/2005), in a twofold dilution series (200 nM–0 nM), were sequentially injected onto the chip surface. For each concentration, the antigen was injected for 5 min (association phase), followed by HBST + BSA running buffer injection for 15 min (dissociation phase). Two regeneration cycles of 15 s were performed after the antigen dilution series by injecting 10 mM glycine pH 2 on the chip surface. Data were analysed using the Carterra Kinetics analysis software in a 1:1 Langmuir binding model to determine association ($k_a$), dissociation ($k_d$), binding affinity constants ($K_D$) and $R_{max}$.

## Virus for lethal challenge

An infectious clone of A/dairy cattle/Texas/24008749001/2024, the first sequenced virus published in Global Initiative on Sharing All Influenza Data (GISAID) from the outbreak in dairy cattle[61,62], was used in mouse lethal challenge studies. Plasmids with both a pol I and pol II expression system (based on the pHW system) were synthesized by Twist Biosciences on the basis of sequences deposited in GISAID (accession no. EPI_ISL_19014384), with non-coding regions for each segment determined from consensus alignment of H5N1 strains from the 2.3.4.4b clade viruses. One plasmid was synthesized for each virus segment (8 in total). The plasmids containing the eight segments were each diluted to a concentration of 100 ng µl$^{-1}$, and a total of 500 ng of each gene segment was combined with 100 µl of Opti-MEM and 5 µl of Lipofectamine 2000 transfection reagent (Life Technologies). The transfection mixture was incubated at room temperature for 25 min and transferred to 293T cells in Opti-MEM complete media (Life Technologies) in a 6-well plate. After 24 h of incubation at 37 °C with 5% CO$_2$, 750,000 Madin–Darby canine kidney (MDCK) cells (ATCC, CCL-34) were added to the 293T cells. Following another 24-h incubation, the supernatant from

the cells was transferred to MDCK cells in a T75 cm$^2$ flask containing MEM medium with L-glutamine and tosyl-phenylalanine chromomethyl ketone (TCPK)-treated trypsin. The flask was monitored for cytopathic effect (CPE) for 48 h post inoculation; once CPE was confirmed, the supernatant was collected and virus titre determined by plaque assay. Aliquots of the rescued reverse genetic virus were then inoculated into 10-day embryonated SPF chicken eggs (AVS Bio) that were incubated at 38 °C for 24 h followed by an overnight incubation at 4 °C before collection of the amniotic fluid. The virus stock was sequenced and the absence of mutations compared to the sequence deposited in GISAID EPI_ISL_19014384 was confirmed.

## Passive transfer and viral challenge

BALB/c female mice (6–10 weeks old, Charles River) were anaesthetized with isoflurane for all procedures. The day before challenge, mice (10 per group) were inoculated with antibodies by intraperitoneal injection (200 µl administered using a 1-ml syringe and 25-gauge needle). Mice were also marked at the same time with ear tags for tracking purposes. For challenge, a dose of 6 plaque-forming units (p.f.u.s) (3× LD$_{50}$) was used (50% lethal dose (LD$_{50}$) was established to be ≤2 p.f.u.s. based on preliminary studies). Dose was confirmed by plaque assay. Virus challenge was by intranasal inoculation, applying 50 µl by micropipette to the nares while mice were anaesthetized. Mice were weighed once daily and checked twice daily for 14 days. Clinical signs of disease (changes in appearance and behaviour, neurological signs) were noted at each check. Mice that were either moribund, suffering respiratory distress, had a ≥20% loss of body weight from baseline, or with severe neurological signs (seizures and/or hindlimb paralysis) were euthanized immediately using carbon dioxide. Euthanasia was performed using procedures consistent with the American Veterinary Medical Association guidelines.

## Plaque assay

Virus stock titre and challenge dose for mouse protection studies were confirmed by plaque assay on MDCK cells (ATCC, CCL-34). Samples were serially diluted and absorbed onto monolayers of MDCK cells in 6-well plates, which were overlaid with agarose-containing EMEM medium and incubated for 3 days at 37 °C and 5% CO$_2$. The wells were then fixed with formaldehyde and stained with 0.25% crystal violet to visualize plaques.

## Pharmacokinetic study in human neonatal Fc receptor-Fc (FcRn-Fc) transgenic mice

Half-life extension LS mutations (M428L/N4324S) were introduced in the IgG1 Fc domain of mAbs[48]. Human FcRn-Fc transgenic mice (mFcRn$^{-/-}$ hFcRn-Fc Tg mice, JAX 029686, The Jackson Laboratory) were used to assess the pharmacokinetics of selected antibodies. Each animal was infused intravenously with 5 mg mAb kg$^{-1}$ of body weight (5 mice per mAb condition). Whole blood samples were collected on days 1, 2, 5, 7, 9, 14, 21, 28, 35, 42, 49 and 56.

All mice were bred and maintained under pathogen-free conditions at the AAALAC-accredited Animal Facility at the National Institute of Allergy and Infectious Diseases and housed in accordance with the procedures outlined in the Guide for the Care and Use of Laboratory Animals. All mice were between 8 and 12 weeks of age and a mix of males and females.

Serum mAb levels were quantified using plates coated with HA (H5/Texas/37/2024). Briefly, Nunc MaxiSorp (Thermo Fisher) plates were coated with 100 ng well$^{-1}$ of HA in PBS, washed with PBS-T five times and blocked with a blocking buffer of Tris-buffered saline-Tween 20 (TBS-T) with 5% milk and 2% bovine serum albumin (BSA) for at least 1 h. Fivefold serial dilutions ranging from 1:20 to 1:12,500 of plasma were made in blocking buffer. For each sample, three consecutive serial dilutions were plated in duplicate. An antibody-specific standard curve (starting at 200 ng ml$^{-1}$ with 8 twofold serial dilutions) and positive, negative and

spiked controls were included on each plate. Samples were incubated on the plate for 1 h at r.t., followed by a PBS-T wash. Bound mAbs were probed with a horseradish peroxidase-labelled donkey anti-human IgG (1:10,000 dilution; Jackson ImmunoResearch) for 30 min at r.t. The plate was washed and 100 µl SureBlue TMB (SeraCare) substrate was added. Once colour was developed (typically 15 to 20 min), stopping buffer (100 µl 1 N $H_2SO_4$) was added and the optical density at 450 nm was read. Softmax Pro v.5.2revC (Molecular Devices) software was used to calculate mAb concentrations. Pharmacokinetic parameters were calculated using Phoenix WinNonlin 8.5.1.3.

### nsEM

Complexes were formed by mixing HA with Fab at a slight molar excess of Fab over HA protomer. Samples were immediately diluted to ~0.02 mg ml$^{-1}$ with buffer containing 10 mM HEPES pH 7.0 and 150 mM NaCl, and applied to a freshly glow-discharged carbon-coated copper grid for ~15 s. After removing excess liquid, the grid was washed twice with the same buffer, and adsorbed proteins were negatively stained by consecutively applying 2 drops of 0.75% uranyl formate. Datasets were recorded at a nominal magnification of ×57,000 (corresponding to a pixel size of 2.53 Å) on a ThermoFisher Talos F200C electron microscope operated at 200 kV and equipped with a Ceta camera. Particles were selected from micrographs using Topaz[63]. Two-dimensional (2D) classification, 3D classification and map refinement were performed in Relion[64]. UCSF Chimera[65] was used for map alignment and visualization.

### Cryo-EM structural analysis

The HA–Fab complexes were prepared by mixing HA and individual Fabs at a molar ratio of 1:1.25 (HA protomer:Fab). DDM detergent (0.085 mM) was added to all mixtures except for the complex with Fab 326-289.74 such that the final HA concentration was ~0.5 mg ml$^{-1}$. The mixtures were incubated at room temperature for 5 min. UltrAuFoil holey grids (R 2/2, Quantifoil) were used for the complexes with Fabs 326-366.26 and 310-1H02, and UltrAuFoil holey grids (R 1.2/1.3, Quantifoil) were used for the complexes with Fabs 326-289.74, 310-7D11 and 310-12D03. The grids were glow discharged for 30 s in atmospheric air using a PELCO easiGlow glow discharger (air pressure: 0.39 mBar, current: 20 mA). Vitrification was performed in an FEI Vitribot Mark IV automatic plunge freezer at 4 °C in a chamber with 95% humidity, blotting time of 2 s and sample volume of 2.7 µl.

All grids were imaged on an FEI Titan Krios G1 electron microscope equipped with an Apollo direct electron detector (Direct Electron). Movies were recorded using a super-resolution pixel size of 0.5 Å pixel$^{-1}$ with exposure time of 2.005 s, 24 frames per movie and a total dose of 40.08 e$^-$ Å$^{-2}$. The datasets were collected using SerialEM 4.09. Detailed data collection statistics are presented in Supplementary Table 1.

Single-particle cryo-EM data analysis was performed using cryoSPARC[66]. Raw movies were pre-processed using patch motion correction and patch CTF estimation jobs. The resulting micrographs were subjected to curation based on full-frame motion, CTF fit resolution and relative ice thickness. Blob picking was used to pick the initial set of particles from a small subset of micrographs. Several rounds of 2D classification were performed to filter out low-quality particles. High-quality 2D class averages were used for template-based particle picking in the entire dataset. The particles were extracted using a box size of 400 × 400 pixels and subjected to several rounds of 2D classification. Three 3D maps were generated from the resulting particles using ab initio reconstruction. These three maps were then subjected to heterogeneous refinement. The resulting best map was then refined separately with C3 symmetry imposed. To aid with model building, the final map was sharpened with 'deepEMhancer'[67]. Details of single-particle analysis are presented in Supplementary Table 1. The initial atomic models of HA and the Fabs were generated using 'ColabFold'[68]. Sections of the models that were well resolved in the sharpened cryo-EM maps

were identified. These sections were isolated and positioned individually in the corresponding regions of the cryo-EM map. The remaining residues were manually built using Coot[69] and ChimeraX[70]. The models were further refined by alternating rounds of real-space refinement in Phenix[71] and model building in Coot and ISOLDE[72]; unsharpened, cryoSPARC-sharpened and DeepEMhancer-sharpened maps were all utilized for model building. The final real-space refinement against the unsharpened map was performed in Phenix. Refinement statistics are summarized in Supplementary Table 1.

### Statistical analysis

Statistical analysis was performed with GraphPad Prism 10.0. Specific details of statistical analysis are indicated in the figure legends and results section. These include the type of statistical test used, $n$ values, and whether the mean, median, s.d. or s.e.m. was calculated and shown. $P$ values equal to or below 0.05 were considered significant. Normality tests were conducted on all data to determine the appropriate statistical test. All statistical tests used are two-tailed.

### Reporting summary

Further information on research design is available in the Nature Portfolio Reporting Summary linked to this article.

## Data availability

The cryo-EM maps and atomic coordinates of the H5N1 TX/24 HA–Fab complexes for mAbs 326-366.26, 310-1H02, 326-289.74, 310-7D11 and 310-12D03 have been deposited in the Electron Microscopy Data Bank (EMDB) and the Protein Data Bank (PDB). The corresponding EMDB accession codes are EMD-48515, EMD-48516, EMD-48517, EMD-48518 and EMD-48521, while the PDB accession codes for the atomic coordinates are 9MQ7, 9MQ8, 9MQ9, 9MQA and 9MQD, respectively. High-throughput SGS data are deposited in the NCBI BioProject under accession PRJNA1220181. All data, analysis and figures related to deep mutational scanning experiments have been archived on Zenodo at https://doi.org/10.5281/zenodo.16740862 (ref. 73). The deep mutational scanning analysis pipeline and data are also publicly available on Github at https://github.com/dms-vep/Flu_H5_American-Wigeon_South-Carolina_2021-H5N1_DMS (ref. 74). Nucleotide sequences for mAbs are in Genbank accession numbers PX104069–PX104338. Requests for materials should be addressed to the corresponding authors. mAbs under patent can be provided with a material transfer agreement. Source data are provided with this paper.

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

## Acknowledgements

We thank R. Koup, G. Alvarado and members of the VRC Influenza Program for helpful discussion; D. Ambrozak and D. Ischiu for help with single-cell sorting; VRC Clinical Trials Program and vaccine trial participants for clinical trial specimens; G. Lee (Zymo) for help with the RNA extraction protocol; and M. Choe and K. Stanczak for help with pharmacokinetics experiments. This study was supported in part by the Intramural Research Program of the National Institutes of Health (NIH) (T.C.P., E.A.B., D.C.D., T.Z., M.K. and S.F.A.); the Frederick National Laboratory for Cancer Research, NIH, under contract HHSN261200800001 (T.M. and Y.T.); R01 AI141707 and 75N93021C00015 (J.D.B.); and NIH contract HHSN261201500003I and NIH/NIAID grant UC7AI180311 (D.S.R.). J.D.B. is an HHMI Investigator. The NIH Office of AIDS Research (OAR) provided Innovation Program funding that supported virus genome sequencing (E.A.B.). The contributions of the NIH authors are considered works of the US Government. The findings and conclusions presented in this paper are those of the authors and do not necessarily reflect the views of the NIH or the US Department of Health and Human Services.

## Author contributions

M.K. and S.F.A. conceptualized the project. A.A.A.-S., G.F., A.C., M.J.V., T.M., G.E.M., R.A.G., V.G.C., B.D., D.S.R., J.D.B., Y.T., E.A.B., T.Z., M.K. and S.F.A. performed formal analysis. A.A.A.-S., G.F., A.C., M.J.V., T.M., G.E.M., G.D.S., R.A.G., V.G.C., B.D., M.D.R., A.J.C., E.B.-L.,T.B., A.R.H., J.R.-T., T.S.J., S.S., E.S.Y., C.C., E.L.W., M.R., D.S.R., J.D.B., Y.T., E.A.B., D.C.D., T.Z., M.K. and S.F.A. conducted investigations. A.C., G.D.S., B.D., I.J.G., T.S.D., L.D. and J.D.B. procured resources. A.A.A.-S., T.Z., M.K. and S.F.A. wrote the original manuscript draft. All authors reviewed and edited the manuscript. A.A.A.-S., T.M., Y.T., T.Z., M.K. and S.F.A. peformed visualization. D.S.R., T.C.P., J.D.B., E.A.B., D.C.D., T.Z., M.K. and S.F.A. supervised the project. D.S.R., T.C.P., J.D.B., E.A.B., D.C.D., T.Z., M.K. and S.F.A. acquired funding.

## Competing interests

A.A.A.-S., G.F., G.E.M., M.K. and S.F.A. are listed as inventors on patent application US 63/781,226 based on the study presented in this paper, which has been filed by the US Department of Health and Human Services. J.D.B. and B.D. are inventors on Fred Hutch licensed patents related to viral deep mutational scanning. J.D.B. consults for Invivyd, Apriori Bio, GSK and Pfizer. The other authors declare no competing interests.

## Additional information

**Extended data** is available for this paper at https://doi.org/10.1038/s41564-025-02137-x.

**Correspondence and requests for materials** should be addressed to Tongqing Zhou, Masaru Kanekiyo or Sarah F. Andrews.

[1]Vaccine Research Center, National Institute of Allergy and Infectious Diseases, National Institutes of Health, Bethesda, MD, USA. [2]Vaccine Research Center Electron Microscopy Unit, Cancer Research Technology Program, Leidos Biomedical Research, Inc., Frederick National Laboratory for Cancer Research, Frederick, MD, USA. [3]Fred Hutchinson Cancer Center, Seattle, WA, USA. [4]Center for Vaccine Research, University of Pittsburgh, Pittsburgh, PA, USA. [5]Howard Hughes Medical Institute, Chevy Chase, MD, USA. [6]These authors contributed equally: Alexandra A. Abu-Shmais, Gray Freeman. ✉e-mail: tzhou@nih.gov; kanekiyom@nih.gov; sarah.andrews2@nih.gov

**Extended Data Table 1 | VRC310 trial regimen and participant demographics**

| Trial Regimen | | Sex | Age | Time Point Sorted | % H5 TX/24 head-specific B cells | # of mAbs expressed |
|---|---|---|---|---|---|---|
| DNA prime | 16 wk MIV boost | Female | 45 | 2 wks p.b. | 0.31% | 7 |
| DNA prime | 4 wk MIV boost | Female | 54 | 2 wks p.b. | 0.18% | 8 |
| DNA prime | 12 wk MIV boost | Female | 21 | 2 wks p.b. | 0.06% | 15 |
| DNA prime | 16 wk MIV boost | Female | 36 | 2 wks p.b. | 0.14% | 9 |
| DNA prime | 16 wk MIV boost | Female | 28 | 2 wks, 6 mos p.b. | 0.14%, 0.11%, | 20 |
| MIV prime | 24 wk MIV boost | Female | 57 | 6 mos p.b. | 0.41% | 23 |
| DNA prime | 12 wk MIV boost | Male | 57 | 6 mos p.b. | 0.08% | 3 |
| DNA prime | 4 wk MIV boost | Female | 31 | 6 mos p.b. | 0.32% | 15 |
| *DNA prime | 4 wk MIV boost | Female | 20 | 6 mos p.b. | 0.21% | 9 |
| DNA prime | 8 wk MIVboost | Female | 29 | 6 mos p.b. | 0.13% | 22 |
| DNA prime | 24 wk MIV boost | Male | 26 | 6 mos p.b. | 0.05% | 4 |

Trial regimen, age, sex, time point sampled, percentage of H5 TX/24 head-specific B cells (of IgG+, IgA+, IgM- B cells), and the number of recombinantly expressed monoclonal antibodies from each of the 11 donors evaluated in this study. Asterisk denotes trial participant enrolled in the FluMos-v2 study.

**Extended Data Table 2 | Characteristics of top 5 mAbs**

| | 310-12D03 | 310-1H02 | 310-7D11 | 310-289.74 | 310-366.26 | MEDI8852 |
|---|---|---|---|---|---|---|
| VH | VH4-61 | VH3-15 | VH3-33 | VH1-69 | VH4-4 | |
| VH SHM (%) | 5.5 | 4.4 | 3.8 | 4.5 | 6.9 | |
| CDRH3 | ATSYITGIQGVDY | TTLYCAITTCFSPRY | ARDGFPNCISATCGYAMDV | ARDQYSNYFYGLDV | ARGLGWEQGGYRAFDM | |
| VK/L | VK3-20 | VL2-23 | VK2-28 | VK3-11 | VK3-15 | |
| VK/L SHM (%) | 1.5 | 2.6 | 2.9 | 2.5 | 2.5 | |
| CDRL3 | QQYGYSLTWT | CSYVGSDTWGV | MQGLETPFT | QQRSNWPPLYS | QQHYNWPPLT | |
| H5 TX/24 IC80 | 0.007 | 0.02 | 0.006 | 0.017 | 0.013 | 0.124 |
| H5 Sich/14 IC80 | 0.001 | 0.006 | 0.013 | 0.006 | 0.008 | 0.036 |
| H5 Cam/23 IC80 | 0.002 | 0.273 | 0.01 | 0.133 | >25 | 0.328 |
| H5 Vict/24 IC80 | 0.749 | 0.067 | >25 | 0.335 | 0.193 | 0.742 |
| H5 Indo/05 IC80 | 0.001 | 0.005 | 0.004 | 0.014 | 0.009 | 0.177 |
| H5 Viet/04 IC80 | >25 | >25 | 0.008 | 0.022 | 0.012 | 0.214 |
| H2 Sing/57 IC80 | >25 | >25 | >25 | >25 | 0.135 | 0.267 |
| H5 Mex/24 IC80 | 0.006 | 0.221 | 1.884 | >25 | >25 | 0.955 |

Sequence characteristics and neutralization IC$_{80}$ values in µg/ml of top H5 TX/24 antibodies matching Fig. 3c. Percent identity is calculated at the nucleotide level.

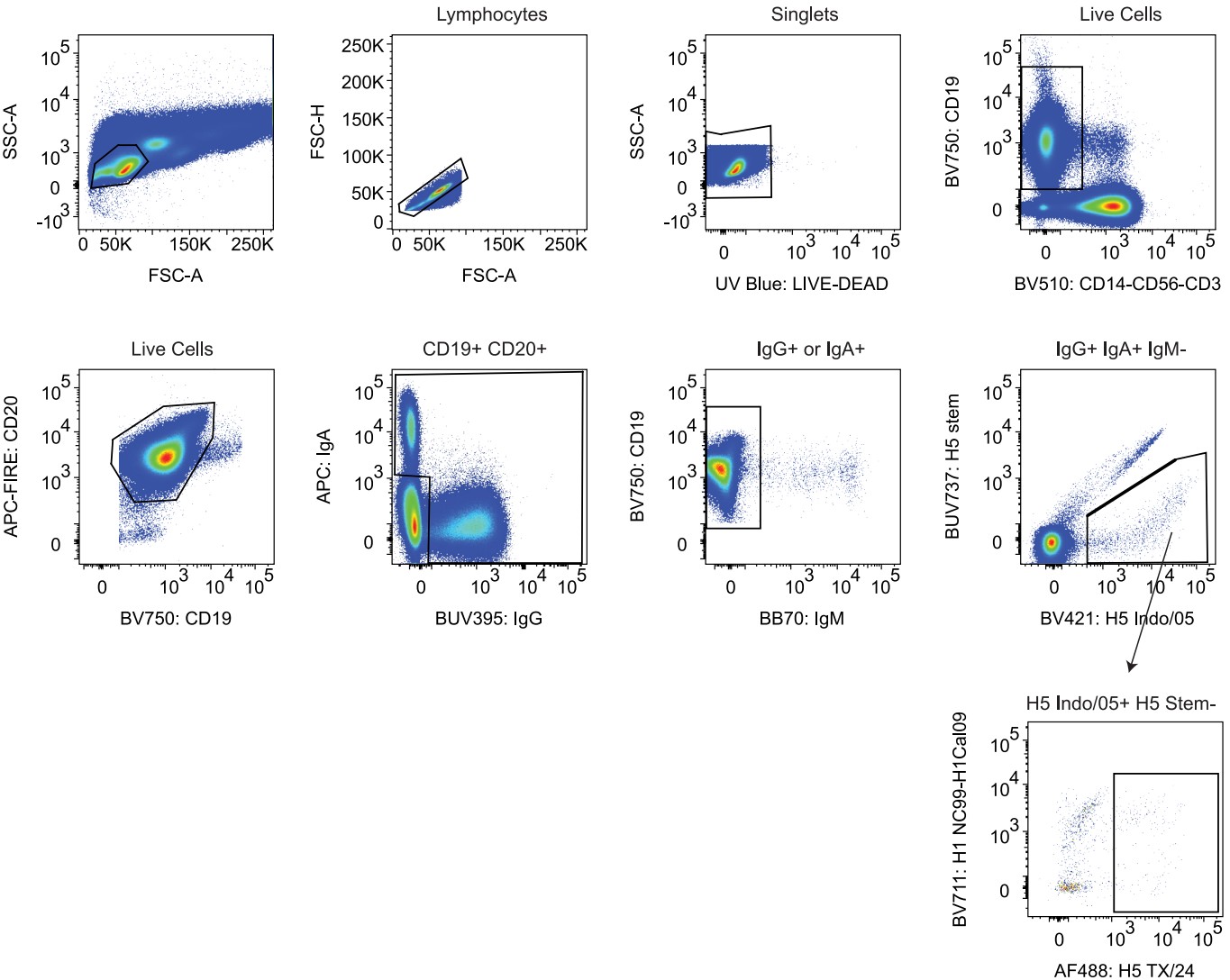

**Extended Data Fig. 1 | Flow cytometry gating for H5 TX/24 specific B cells.** Representative flow plots for isolation of H5 TX/24 + B cells. PBMCs isolated at peak time points were stained with antibodies to detect B cell surface markers and fluorescently labeled HA probes. CD19 + CD20+IgG+ or IgA+ were gated and reactivity of B cells to the ectodomain of H5 Indo/05 and H5 TX/24, but not H5 Indo/05 stabilized stem, were single cell sorted.

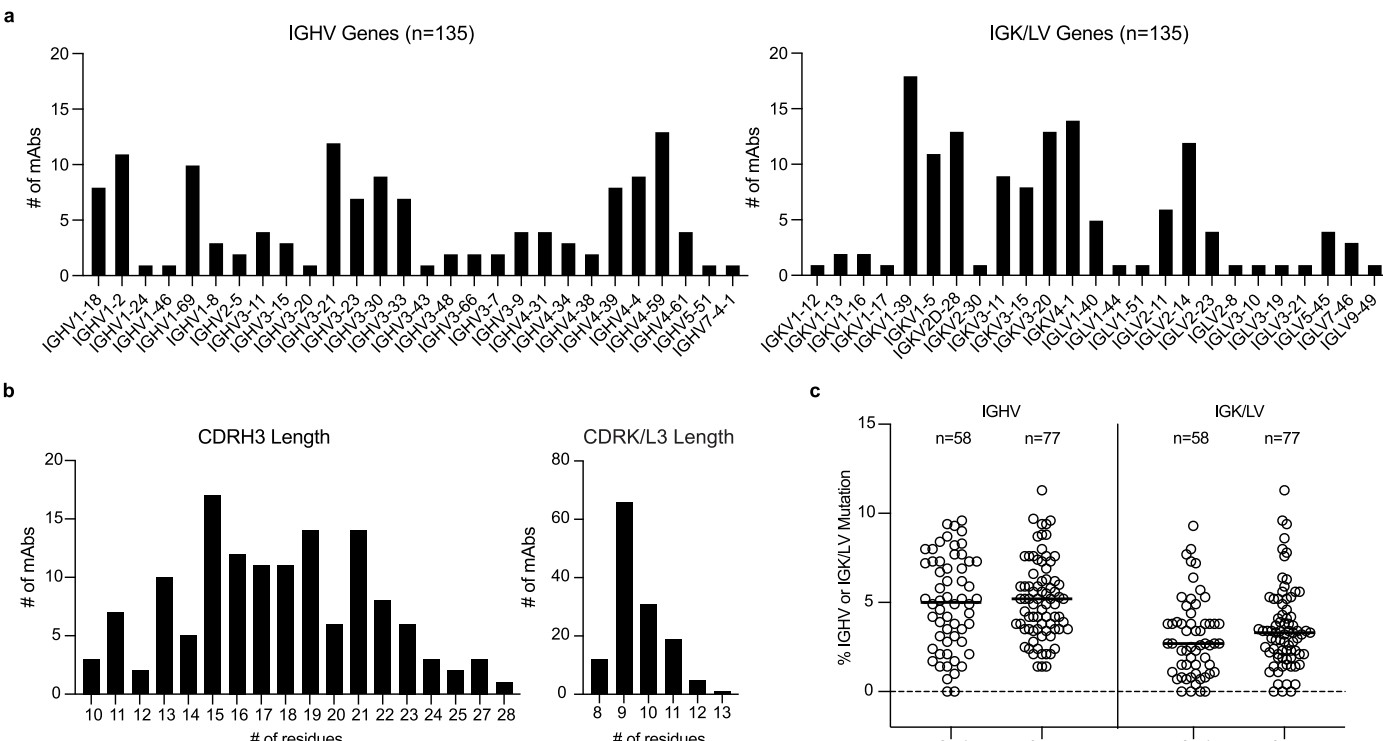

**Extended Data Fig. 2 | Genetic characterization of H5-specific mAbs. a**, Number of clonally distinct H5-specific mAbs characterized in Fig. 2 encoded by heavy (left) and light (right) variable genes as indicated. **b**, Number of mAbs with the indicated CDRH3 (left) or CDRK/L3 (right) length. **c**, Percent immunoglobulin heavy or light variable chain SHM of mAbs isolated at 2wks or 6mos. There was no significant difference between the two time points by two-tailed Mann-Whitney test.

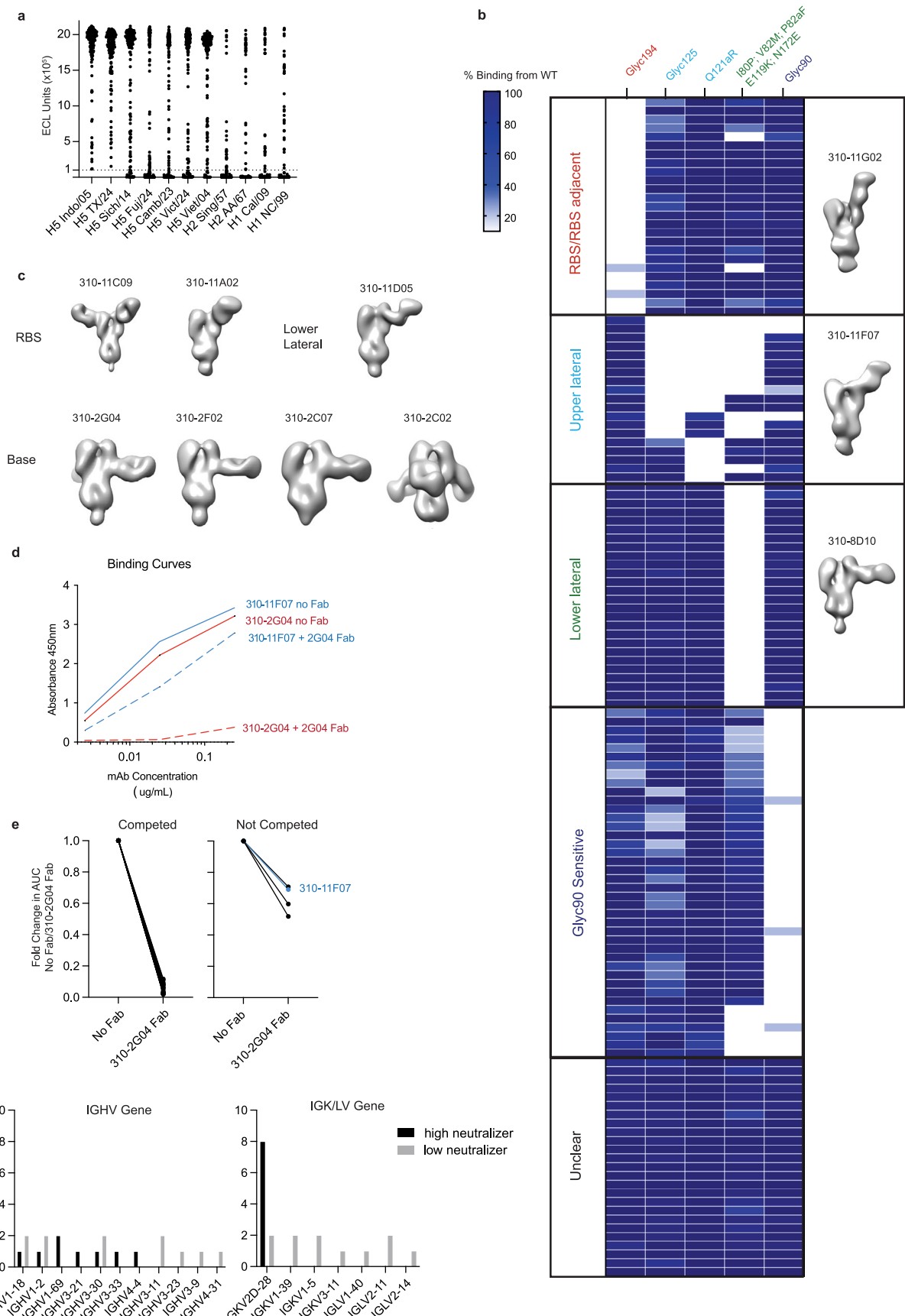

**Extended Data Fig. 3 | See next page for caption.**

**Extended Data Fig. 3 | Epitope characterization of mAbs. a**, Binding AUC of mAbs to indicated HAs. Each dot is one mAb. The dotted line indicates the threshold for positive binding. **b**, Percent binding of each mAb in each row to each indicated H5 TX/24 HA mutant relative to wildtype. To the right of the heatmap are nsEM of representative Fabs for each category in complex with HA. We were unable to obtain nsEM images from the Glyc90 sensitive group. The unclear mAbs had little to no reduction in binding to mutants. **c**, nsEM of Fabs in complex with HA of 7 of the 25 unclear mAbs demonstrating targeting to the RBS, lower lateral and base region. **d,e**, Fabs of base binding 310-2G04 were used in a competition ELISA with the remaining 18 unclear mAbs. In d is shown

representative binding curves of 310-2G04 and upper lateral mAb 310-11F07 as a control with or without addition of 310-2G04 Fab. In e is indicated the fold change in binding AUC of each mAb with or without 310-2G04 Fab competition for binding. 310-2G04 Fab competed for binding to H5 TX/24 for 16 of the unclear mAbs indicating they bound the base region (left graph). Two mAbs did not have a large reduction in binding in the presence of 310-2G04 (right graph) and their epitope remained unclear. **f**, IGHV and IGK/LV gene usage of the upper lateral epitope targeting mAbs categorized as high neutralizers (black) or low neutralizers (gray).

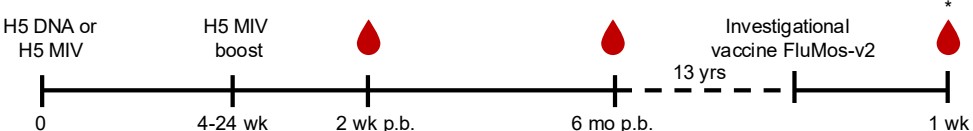

**Extended Data Fig. 4 | Isolation of two antibodies after an investigational vaccine.** Two monoclonal antibodies, 326-289.74 and 326-366.26, were identified from a participant of both the H5N1 and a subsequent vaccine trial. Plasmablasts were collected one week after the participant received investigational vaccine, FluMos-v2, as part of an ongoing Phase I clinical trial VRC 326 (NCT05968989). Asterisk denotes timepoint from which the mAbs were isolated.

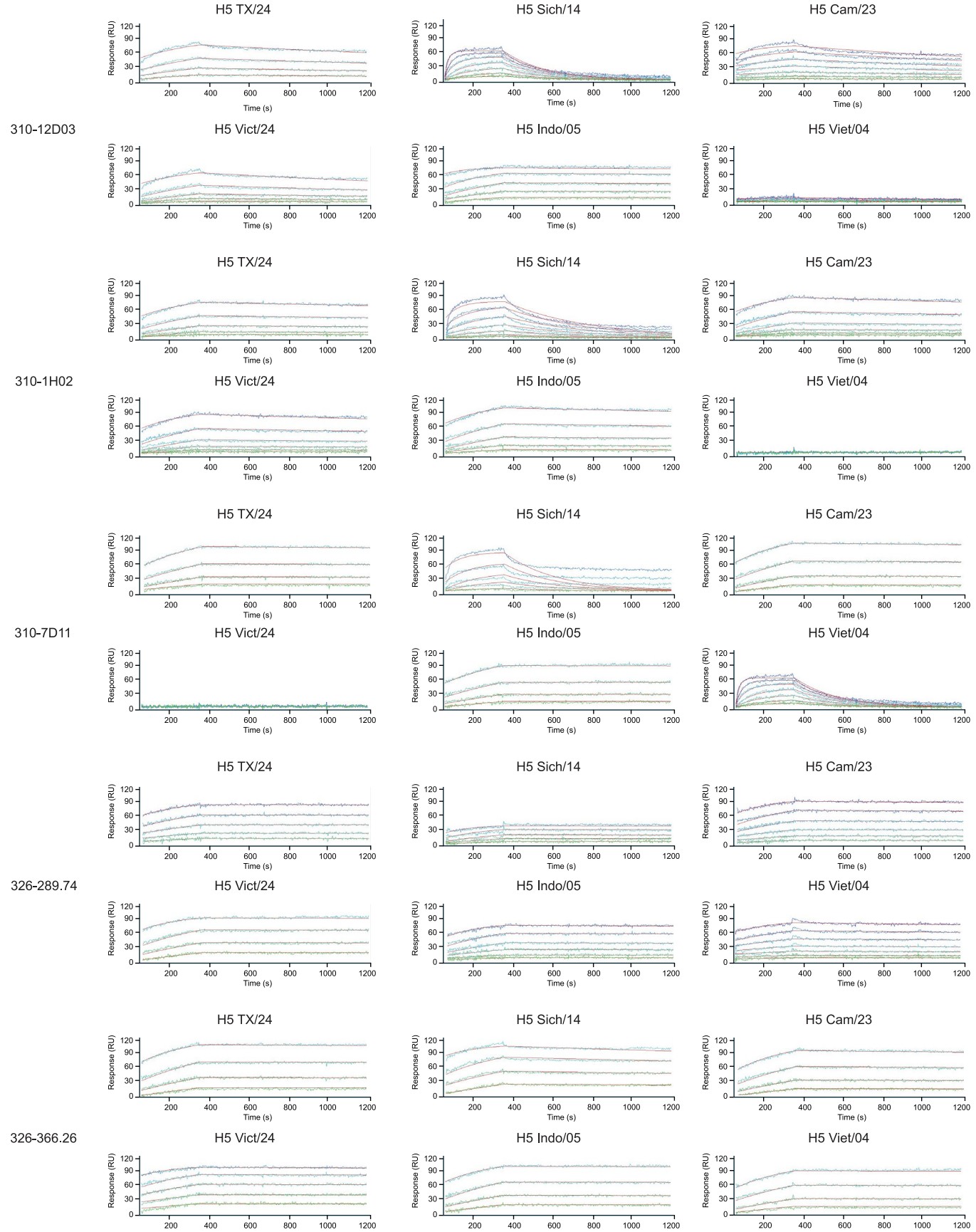

**Extended Data Fig. 5 | Surface Plasmon Resonance binding curves.** Association (ka) and dissociation curves (kd) of Fabs to immobilized trimeric HA.

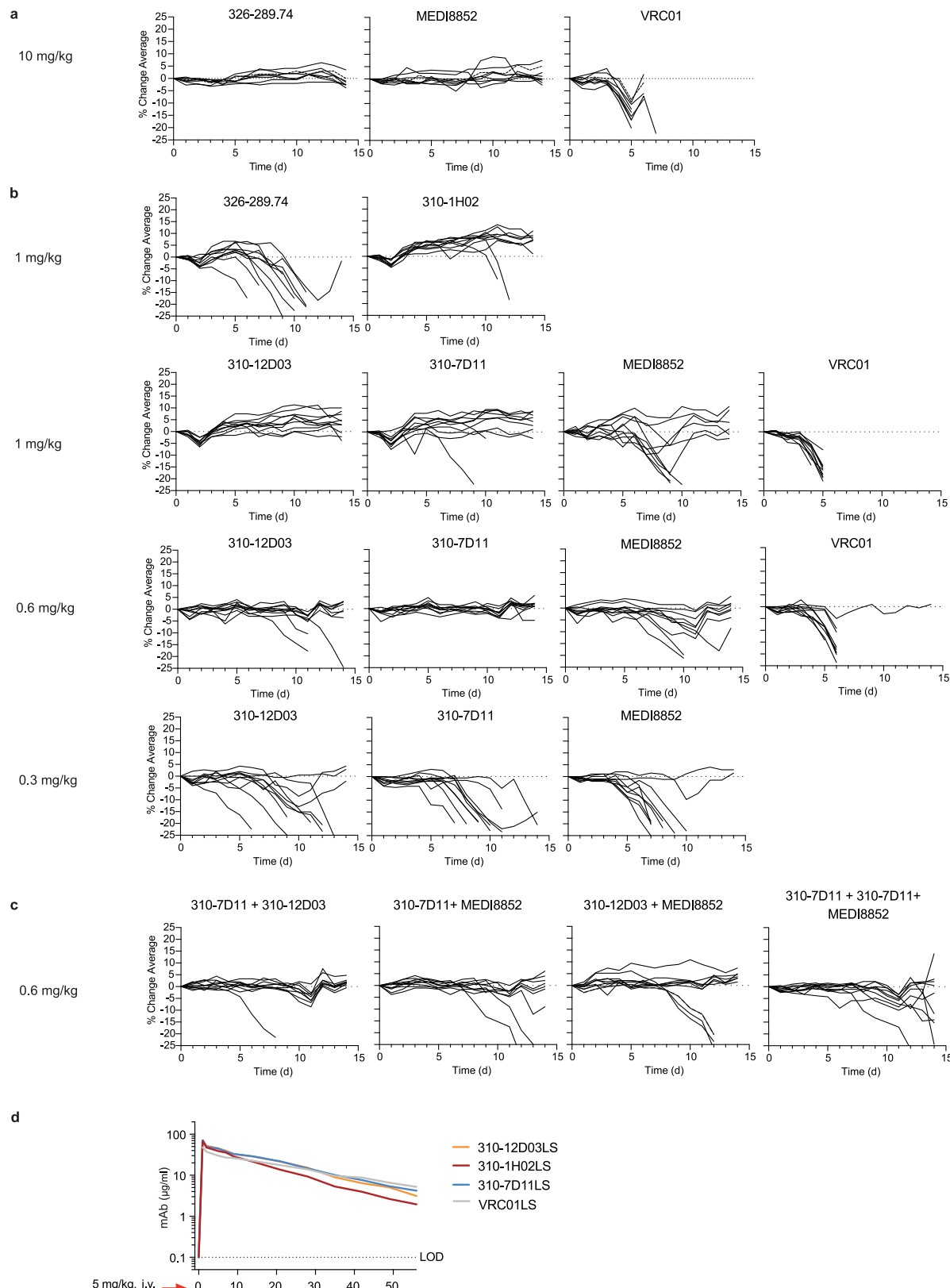

**Extended Data Fig. 6 | Weight loss curves and pharmacokinetics.**
**a-c**, Weight loss curves for individual mice in mAb treatment groups at indicated concentrations. **d**, Pharmacokinetics of top 3 mAbs and VRC01 with LS mutations in human FcRn knock-in mice after an IV dose of 5 mg/kg. Shown is the average serum mAb concentration of 5 animals/mAb, reaching a concentration maximum ($C_{max}$) of 69-74 μg/ml with an estimated serum half-life of 16 days (310-12D03 and 310-7D11), and 13 days (310-1H02).

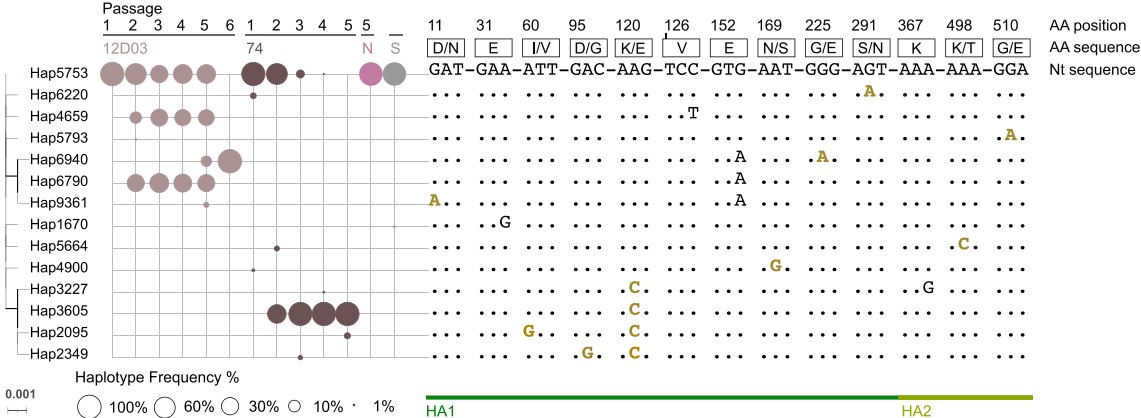

**Extended Data Fig. 7 | Haplotype Distribution of Passages and Stock of Hemagglutinin.** Phylogenetic relationships of hemagglutinin haplotypes derived from passages and stock, along with their corresponding Log10 frequency percentages. The tree was constructed using the maximum likelihood method with 1,000 bootstrap replicates, and branches with a bootstrap support of 65 or higher are indicated with thicker lines. The labels indicate the following: N = passages without mAb, S = stock, 12D03 = passages with 310-12D03, 74 = passages with 326-289.84. The alignment highlights nucleotide changes, with positions annotated using the H3 codon annotation.

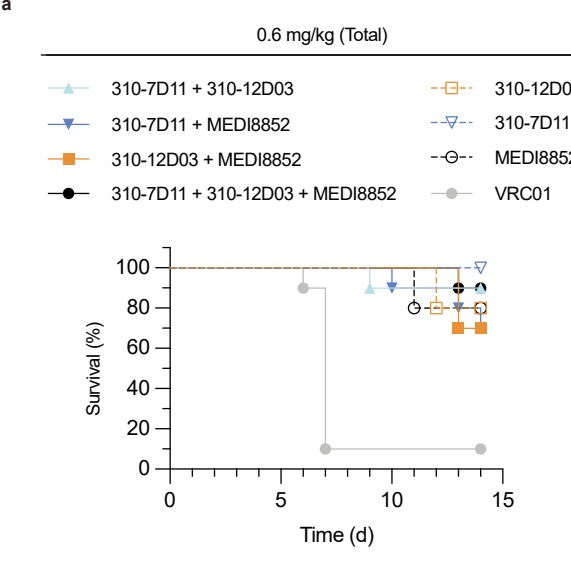

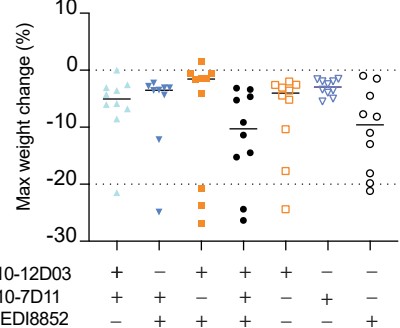

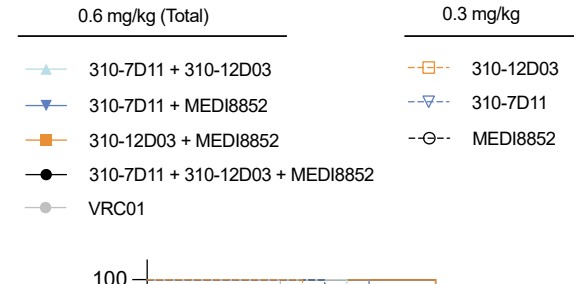

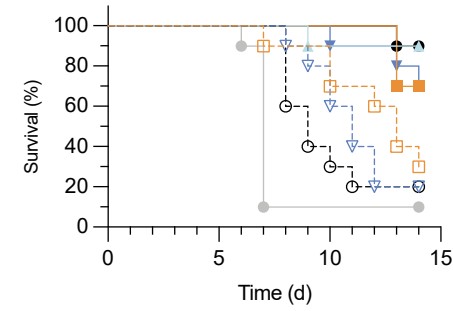

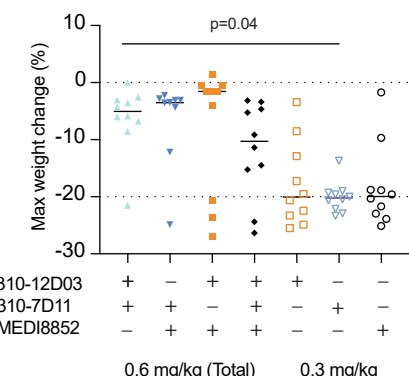

**Extended Data Fig. 8 | *In vivo* protection comparing combinations and individual mAbs. a, b**, Protection and maximum weight change (%) upon H5 TX/24 challenge after passive transfer of combinations of mAbs (total 0.6 mg/kg) and individual mAbs at 0.6 mg/kg (a) or 0.3 mg/kg (b). Data shown for individual mAbs is same as in Fig. 3. Statistical significance of survival was determined by log-rank (Mantel-Cox) test with Bonferroni-Sidak adjustment. All mAb combinations were statistically significant relative to VRC01 (p < 0.001); p = 0.0002 for 310-7D11 + 310-12D03, p = 0.0006 for 310-12D03 + MEDI8852, p = 0.0008 for 310-7D11 + MEDI8852, p = 0.0002 for 310-7D11 + 310-12D03 + MEDI8852. There was no significant difference in survival between

combinations and individual mAbs at 0.6 mg/kg. All combinations except 310-12D03 + MEDI88252 were statistically more protective than individual mAbs at 0.3 mg/kg. For comparisons against 310-7D11 0.3 mg/kg, p = 0.0032 for 310-7D11 + 310-12D03, p = 0.0102 for 310-7D11 + MEDI8852, p = 0.0008 for 310-7D11 + 310-12D03 + MEDI8852. For comparisons against 310-12D03 0.3 mg/kg, p = 0.011 for 310-7D11 + 310-12D03 and p = 0.0055 for 310-7D11 + 310-12D03 + MEDI8852. For comparisons against MEDI8852 0.3 mg/kg, p = 0.0015 for 310-7D11 + 310-12D03, p = 0.0058 for 310-7D11 + MEDI8852, p = 0.0028 for 310-12D03 + MEDI8852, and p = 0.0007 for 310-7D11 + 310-12D03 + MEDI8852.

# Reporting Summary

## Statistics

For all statistical analyses, confirm that the following items are present in the figure legend, table legend, main text, or Methods section.

| n/a | Confirmed | |
|---|---|---|
| ☐ | ☒ | The exact sample size ($n$) for each experimental group/condition, given as a discrete number and unit of measurement |
| ☐ | ☒ | A statement on whether measurements were taken from distinct samples or whether the same sample was measured repeatedly |
| ☐ | ☒ | The statistical test(s) used AND whether they are one- or two-sided *Only common tests should be described solely by name; describe more complex techniques in the Methods section.* |
| ☒ | ☐ | A description of all covariates tested |
| ☐ | ☒ | A description of any assumptions or corrections, such as tests of normality and adjustment for multiple comparisons |
| ☐ | ☒ | A full description of the statistical parameters including central tendency (e.g. means) or other basic estimates (e.g. regression coefficient) AND variation (e.g. standard deviation) or associated estimates of uncertainty (e.g. confidence intervals) |
| ☐ | ☒ | For null hypothesis testing, the test statistic (e.g. $F$, $t$, $r$) with confidence intervals, effect sizes, degrees of freedom and $P$ value noted *Give P values as exact values whenever suitable.* |
| ☒ | ☐ | For Bayesian analysis, information on the choice of priors and Markov chain Monte Carlo settings |
| ☒ | ☐ | For hierarchical and complex designs, identification of the appropriate level for tests and full reporting of outcomes |
| ☒ | ☐ | Estimates of effect sizes (e.g. Cohen's $d$, Pearson's $r$), indicating how they were calculated |

*Our web collection on statistics for biologists contains articles on many of the points above.*

## Software and code

Policy information about availability of computer code

| | |
|---|---|
| Data collection | FACSDiva v8<br>SerialEM 4.09 |
| Data analysis | FlowJo v10<br>bclfastq v2.20.0.422<br>SONAR v4.3<br>MEGA11<br>Skylign<br>GraphPad Prism v10<br>polyclonal package (https://jbloomlab.github.io/polyclonal/)<br>Perl v5.16.3<br>RStudio v2022.07.1<br>Minimap2<br>SeqKit v2.3.1<br>Cutadapt v4.0<br>MAFFT v7.467<br>Geneious Prime v2023.0.4<br>Carterra Kinetics<br>Topaz<br>Relion<br>cryoSPARC |

deepEMhancer
ColabFold
Coot
Phenix
ISOLDE
UCSF ChimeraX

For manuscripts utilizing custom algorithms or software that are central to the research but not yet described in published literature, software must be made available to editors and reviewers. We strongly encourage code deposition in a community repository (e.g. GitHub). See the Nature Portfolio guidelines for submitting code & software for further information.

## Data

Policy information about availability of data

All manuscripts must include a data availability statement. This statement should provide the following information, where applicable:
- Accession codes, unique identifiers, or web links for publicly available datasets
- A description of any restrictions on data availability
- For clinical datasets or third party data, please ensure that the statement adheres to our policy

The cryo-EM maps and atomic coordinates of the H5N1 TX/24 HA –Fab complexes for mAbs 326-366.26, 310-1H02, 326-289.74, 310-7D11, and 310-12D03 have been deposited in the Electron Microscopy Data Bank (EMDB) and the Protein Data Bank (PDB). The corresponding EMDB accession codes are EMD-48515, EMD-48516, EMD-48517, EMD-48518, and EMD-48521, while the PDB accession codes for the atomic coordinates are 9MQ7, 9MQ8, 9MQ9, 9MQA, and 9MQD, respectively. High-throughput SGS data are deposited to NCBI BioProject under accession PRJNA1220181. All data, analysis and figures related to deep mutational scanning experiments have been archive on Zenodo under DOI:10.5281/zenodo.1674086276. Deep mutational scanning analysis pipeline and data is also publicly available on Github at https://github.com/dms-vep/Flu_H5_American-Wigeon_South-Carolina_2021-H5N1_DMS. Nucleotide sequences for mAbs are in Genbank accession numbers PX104069-PX104338. Requests for materials should be addressed to the corresponding authors. mAbs under patent can be provided with a material transfer agreement.

## Research involving human participants, their data, or biological material

Policy information about studies with human participants or human data. See also policy information about sex, gender (identity/presentation), and sexual orientation and race, ethnicity and racism.

| | |
|---|---|
| Reporting on sex and gender | Sex information is available in Extended Data Table 1. |
| Reporting on race, ethnicity, or other socially relevant groupings | Race and ethnicity was collected as part of the original Phase I clinical trial from which samples were obtained, but is not reported here as is irrelevant to this post-hoc study |
| Population characteristics | For the H5 vaccine trial healthy adults aged 18 to 60 years were enrolled. In the FluMos-v2 vaccine trial healthy adults born between 18 and 50 were enrolled. Inclusion criteria for both trials required general good health determined by laboratory tests, medical history, and physical exam. |
| Recruitment | Volunteers were recruited from the greater Washington, DC, area by IRB-approved written and electronic media. |
| Ethics oversight | The trial protocol was reviewed and approved by the NIAID Institutional Review Board. U.S. Department of Health and Human Services guidelines for conducting clinical research were followed. |

Note that full information on the approval of the study protocol must also be provided in the manuscript.

# Field-specific reporting

Please select the one below that is the best fit for your research. If you are not sure, read the appropriate sections before making your selection.

☒ Life sciences ☐ Behavioural & social sciences ☐ Ecological, evolutionary & environmental sciences

For a reference copy of the document with all sections, see nature.com/documents/nr-reporting-summary-flat.pdf

# Life sciences study design

All studies must disclose on these points even when the disclosure is negative.

| | |
|---|---|
| Sample size | Sample sizes of animal studies were determined based on prior experience with similar experiments. Assuming variance in the lethality is proportional to mean for a given group (constant CV of 30%, typical for this type of experiments), a group size of 10 will give 89% power to detect 2-fold differences or a 49% power to detect 1.5-fold differences between groups based on a two tailed test of means with an alpha set to 0.05 (calculation was performed by 1-way ANOVA pairwise tools at powerandsamplesize.com). |
| Data exclusions | No data were excluded. |
| Replication | Structural determination of HA-Fab complexes was performed once as is custom for CryoEM. All in vitro assays were repeated at least twice |

| Replication | with similar results. Animal infection studies were performed once as they are BSL-3 and resource intensive. |
|---|---|
| Randomization | All animals used in the study were randomly assigned to different experimental groups. |
| Blinding | All in vitro experiments were not performed blindly as samples were labeled with mAb names.  However, data collected is quantitative measured by instrumentation, so not open to interpretation or require judgement calls. All in vivo experiments were performed blindly to animal handlers. |

# Reporting for specific materials, systems and methods

We require information from authors about some types of materials, experimental systems and methods used in many studies. Here, indicate whether each material, system or method listed is relevant to your study. If you are not sure if a list item applies to your research, read the appropriate section before selecting a response.

### Materials & experimental systems

| n/a | Involved in the study |
|---|---|
| ☐ | ☒ Antibodies |
| ☐ | ☒ Eukaryotic cell lines |
| ☒ | ☐ Palaeontology and archaeology |
| ☐ | ☒ Animals and other organisms |
| ☐ | ☒ Clinical data |
| ☒ | ☐ Dual use research of concern |
| ☒ | ☐ Plants |

### Methods

| n/a | Involved in the study |
|---|---|
| ☒ | ☐ ChIP-seq |
| ☐ | ☒ Flow cytometry |
| ☒ | ☐ MRI-based neuroimaging |

## Antibodies

| Antibodies used | CD19 BV750, BD#747161, clone SJ25-C1, 1:400 dilution<br>IgG BUV395, BD#564229, clone G18-145, 1:200 dilution<br>IgM BB700, custom, BD, clone G20-127, 1:400 dilution<br>CD3 BV510, Biolegend#317332, clone OKT3, 1:400 dilution<br>CD14 BV510, Biolgend#301842, clone M5E2, 1:200 dilution<br>CD56 BV510, Biolegend# 318340, clone HCD56, 1:200 dilution<br>CD20 APC-Cy7, Biolegend#302313, clone 2H7, 1:400 dilution<br>CD27 BV605, Biolegend#302830, clone O323, 1:100 dilution<br>CD21 PE594, BD#563474, clone B-ly4, 1:400 dilution<br>IgA APC, Miltenyi #130-113-472, clone IS11-8E10, 1:400 dilution |
|---|---|
| Validation | The technical data sheets from the manufacturer for all antibodies state that they are specifically tested and validated to bind the human antigen listed either through testing on human PBMCs or cell lines expressing the antigen of interest.   All antibodies were titered to determine optimal amounts to maximize signal to noise. |

## Eukaryotic cell lines

Policy information about cell lines and Sex and Gender in Research

| Cell line source(s) | Expi293 (ThermoFisher A14527),MDCK-SIAT1-PB1 (Creanga et al. Nat Commun. 2021), 293T (ATCC), MDCK (ATCC, CCL-34) |
|---|---|
| Authentication | Cell lines were not authenticated. |
| Mycoplasma contamination | Tested negative (monthly). |
| Commonly misidentified lines<br>(See ICLAC register) | n/a |

## Animals and other research organisms

Policy information about studies involving animals; ARRIVE guidelines recommended for reporting animal research, and Sex and Gender in Research

| Laboratory animals | BALB/c mice |
|---|---|
| Wild animals | n/a |

| Reporting on sex | Passive transfer study used only female mice because of the previous LD50 titration studies were performed by using female mice. Pharmacokinetics used a mixture of male and female mice. |
| --- | --- |
| Field-collected samples | n/a |
| Ethics oversight | All experiments were conducted in accordance with the National Institutes of Health (NIH) recommendations in the Guide for the Care and Use of Laboratory Animals with pre-approval of specific procedures and protocols by the Institutional Animal Care and Use Committee of the University of Pittsburgh, with animal care in accordance with the Association for Assessment and Accreditation of Laboratory Animal Care (AAALAC). All mice in this study were housed in AAALAC accredited animal facilities in a 12-hour light/dark cycle at an ambient temperature of 22.2 ± 2.8°C with a relative humidity maintained between 30–70%. |

Note that full information on the approval of the study protocol must also be provided in the manuscript.

# Clinical data

Policy information about clinical studies

All manuscripts should comply with the ICMJE guidelines for publication of clinical research and a completed CONSORT checklist must be included with all submissions.

| Clinical trial registration | NCT01086657 and NCT05968989 |
| --- | --- |
| Study protocol | Study details can be found under clinical trial registrations noted above. |
| Data collection | N/A - this is post-hoc use of study samples, not report of clinical study results |
| Outcomes | N/A - this is post-hoc use of study samples, not report of clinical study results |

# Plants

| Seed stocks | *Report on the source of all seed stocks or other plant material used. If applicable, state the seed stock centre and catalogue number. If plant specimens were collected from the field, describe the collection location, date and sampling procedures.* |
| --- | --- |
| Novel plant genotypes | *Describe the methods by which all novel plant genotypes were produced. This includes those generated by transgenic approaches, gene editing, chemical/radiation-based mutagenesis and hybridization. For transgenic lines, describe the transformation method, the number of independent lines analyzed and the generation upon which experiments were performed. For gene-edited lines, describe the editor used, the endogenous sequence targeted for editing, the targeting guide RNA sequence (if applicable) and how the editor was applied.* |
| Authentication | *Describe any authentication procedures for each seed stock used or novel genotype generated. Describe any experiments used to assess the effect of a mutation and, where applicable, how potential secondary effects (e.g. second site T-DNA insertions, mosiacism, off-target gene editing) were examined.* |

# Flow Cytometry

## Plots

Confirm that:

☒ The axis labels state the marker and fluorochrome used (e.g. CD4-FITC).

☒ The axis scales are clearly visible. Include numbers along axes only for bottom left plot of group (a 'group' is an analysis of identical markers).

☒ All plots are contour plots with outliers or pseudocolor plots.

☒ A numerical value for number of cells or percentage (with statistics) is provided.

## Methodology

| Sample preparation | All samples were viably frozen PBMCs thawed at 37C and stained immediately with cell surface markers for analysis and sorting. Cell viability was always above 95%. |
| --- | --- |
| Instrument | All data was collected on a BD FACSymphony S6 |
| Software | Data collection was performed with BD FACSDiva v.8, data analysis was done with FlowJo v.10 |
| Cell population abundance | All sorting was single-cell into individual wells in 96-well plates. Confirmation that B cells were sorted was the ability to amplify immunoglobulin sequences from each well. A subset of immunoglobulin sequences were used to produce monoclonal antibodies that validated specificity. |

Gating strategy | Provided as Extended Data Figure 1

☒ Tick this box to confirm that a figure exemplifying the gating strategy is provided in the Supplementary Information.

