## [Peer Review File · Nature Microbiology]

Cross-neutralizing and potent human monoclonal antibodies against historical and emerging H5Nx influenza viruses

Corresponding Author: Dr Sarah Andrews

Version 0:

Reviewer comments:

Reviewer #1

(Remarks to the Author)

I thank the authors for addressing the review comments by providing more details, insights and additional discussion. I have no additional comments.

Reviewer #2

(Remarks to the Author)

The authors have satisfactorily addressed all of the points in my previous review. I have no further comments/concerns. However, all five PDB cryo-EM structure validation files attached are marked as "This wwPDB validation report is NOT for manuscript review". Across these five preliminary validation reports, within section 8.2 "Resolution estimates", the resolution values from deposited half-maps intersecting FSC 0.143 seem to differ from the reported values by more than 10%. Authors should address these concerns from PDB validation if they have not already done so and provide validated reports.

Reviewer #3

(Remarks to the Author)

Reviewer #4 is now satisfied with the revised manuscript and figures.

Decision Letter:

Our ref: NMICROBIOL-25062187-T

22nd July 2025

Dear Dr. Andrews,

Thank you for submitting your revised manuscript "Isolation of potently neutralizing and protective monoclonal antibodies against H5Nx influenza viruses from individuals immunized with an H5N1 vaccine" (NMICROBIOL-25062187-T). It has now been seen by the original referees and their comments are below. The reviewers find that the paper has improved in revision, and therefore we'll be happy in principle to publish it in Nature Microbiology, pending minor revisions to satisfy the referees' final requests and to comply with our editorial and formatting guidelines.

Thank you again for your interest in Nature Microbiology Please do not hesitate to contact me if you have any questions.

Sincerely,

Reviewer #1 (Remarks to the Author):

The authors have satisfactorily addressed all of the points in my previous review. I have no further comments/concerns.

However, all five PDB cryo-EM structure validation files attached are marked as “This wwPDB validation report is NOT for manuscript review”. Across these five preliminary validation reports, within section 8.2 “Resolution estimates”, the resolution values from deposited half-maps intersecting FSC 0.143 seem to differ from the reported values by more than 10%. Authors should address these concerns from PDB validation if they have not already done so and provide validated reports.

Reviewer #2 (Remarks to the Author):

I thank the authors for addressing the review comments by providing more details, insights and additional discussion. I have no additional comments.

Reviewer #4 (Remarks to the Author):

Reviewer #4 is now satisfied with the revised manuscript and figures.

Version 1:

Decision Letter:

29th August 2025

Dear Dr Andrews,

I am pleased to accept your Article "Cross-neutralizing and potent human monoclonal antibodies against historical and emerging H5Nx influenza viruses" for publication in Nature Microbiology. Thank you for having chosen to submit your work to us and many congratulations.

Authors may need to take specific actions to achieve compliance with funder and institutional open access mandates. If your research is supported by a funder that requires immediate open access (e.g. according to [a href="https://www.springernature.com/gp/open-science/plan-s-compliance"> Plan S principles](https://www.springernature.com/gp/open-science/plan-s-compliance) or the [a href="https://www.springernature.com/gp/open-science/us-federal-agency-compliance"> NIH public access policy](https://www.springernature.com/gp/open-science/us-federal-agency-compliance)) then you should select the gold OA route, and we will direct you to the compliant route where possible. Because authors warrant under our subscription licensing terms that they haven't committed to licensing any version of their article under a licence inconsistent with the terms of our agreement – including the applicable embargo period – publication under the subscription model isn't suitable for authors whose funders require no embargo.

An online order form for reprints of your paper is available at [a href="https://www.nature.com/reprints/author-reprints.html">https://www.nature.com/reprints/author-reprints.html](https://www.nature.com/reprints/author-reprints.html). All co-authors, authors' institutions and authors' funding

agencies can order reprints using the form appropriate to their geographical region.

With kind regards,

P.S. Click on the following link if you would like to recommend Nature Microbiology to your librarian
<http://www.nature.com/subscriptions/recommend.html#forms>

** Visit the Springer Nature Editorial and Publishing website at http://editorial-jobs.springernature.com?utm_source=ejP_NMicro_email&utm_medium=ejP_NMicro_email&utm_campaign=ejp_NMicro for more information about our career opportunities. If you have any questions please click [here](mailto:editorial.publishing.jobs@springernature.com).

Response to Reviewers

Reviewer #1 (Remarks to the Author):

The authors have satisfactorily addressed all of the points in my previous review. I have no further comments/concerns. However, all five PDB cryo-EM structure validation files attached are marked as “This wwPDB validation report is NOT for manuscript review”. Across these five preliminary validation reports, within section 8.2 “Resolution estimates”, the resolution values from deposited half-maps intersecting FSC 0.143 seem to differ from the reported values by more than 10%. Authors should address these concerns from PDB validation if they have not already done so and provide validated reports.

We are now submitting the final PDB cryo-EM structure validation files.

The resolution estimated by the PDB validation in section 8.2 is derived from FSC curve calculated from the unmasked half maps. This essentially corresponds to the 'no mask' curve in Supplementary Figs 1-5, Panel d as calculated during refinement by the CryoSPARC program. Without masking both the protein and solvent are included which results in lower reported resolution. The final resolution is estimated from a corrected FSC curve that applies a mask by thresholding and padding the final volume with high-resolution noise substitution [Chen, S. et al. High-resolution noise substitution to measure overfitting and validate resolution in 3D structure determination by single particle electron cryomicroscopy. Ultramicroscopy 135, 24–35 (2013)]. The mask removes noise from the regions of the box that don't correspond to protein structure resulting in the higher reported resolution.

Reviewer #2 (Remarks to the Author):

I thank the authors for addressing the review comments by providing more details, insights and additional discussion. I have no additional comments.

Reviewer #4 (Remarks to the Author):

Reviewer #4 is now satisfied with the revised manuscript and figures.